# REWARD MODEL BOOSTING FOR RLHF

## ABSTRACT

Reinforcement Learning from Human Feedback (RLHF) is a powerful technique for aligning large language models (LLMs) with human preference. However, it often suffers from the reward hacking issue, where policy optimization improves the proxy reward model while actually degrading performance with respect to the true human preference, due to the imperfection of the proxy. To address this, we propose Reward Model Boosting (RMB), a novel approach that enhances the robustness and reliability of the reward signal for RLHF. RMB first trains a set of reward models with a diverse-promoting regularizer. This encourages each model to learn complementary aspects of the reward landscape. Then, RMB learns a lightweight aggregator in the principle of boosting to aggregate the outputs of the diverse reward models into a more accurate and robust reward signal. Our extensive experiments demonstrate that RMB significantly improves reward accuracy on both in-distribution and out-of-distribution datasets, substantially mitigating the reward hacking issue and ultimately improving RLHF performance.[1]

## 1 INTRODUCTION

LLMs can be guided to follow human preferences through RLHF, which utilizes a reward model trained on labeled human preference data to approximate human judgments of output quality (Bai et al., 2022; Ouyang et al., 2022; Achiam et al., 2023; Touvron et al., 2023). By optimizing the LLM (i.e., the policy) towards this reward model, we can significantly improve the performance of the outputs. This approach has been widely adopted in recent LLM development and has demonstrated strong empirical success in aligning model behavior with human preferences (Ouyang et al., 2022; Gemma et al., 2024; Qwen et al., 2025).

However, a learned reward model for RLHF often suffers from an issue known as *overoptimization* or *reward hacking* (Stiennon et al., 2020; Gao et al., 2023; Coste et al., 2024), where policy optimization improves the proxy reward model while actually degrading performance with respect to the true human preference. To address this, several lines of research have emerged, focusing either on improving the Reinforcement Learning (RL) training process via constrained policy optimization (Moskovitz et al., 2024; Zhang et al., 2024; Liu et al., 2024), or on designing more robust reward models that provide a reliable learning signal for RL training (Coste et al., 2024; Yang et al., 2024; Liu et al., 2025). Our work focuses on the latter direction.

Reward hacking in RLHF arises primarily because a reward model trained on limited human-preference data is inevitably an imperfect surrogate for the true underlying human preference (Gao et al., 2023). A natural way to mitigate this problem is to learn a reward function that is both more robust and more generalizable. For example, Yang et al. (2024) augment reward model training with an auxiliary loss regularizing text generation, which can improve robustness under noisy supervision; however, this auxiliary objective competes with the core preference-learning objective and yields only modest gains. Likewise, Coste et al. (2024) averages the predictions of multiple reward models to reduce individual noise, but different reward models often share the same blind spots and tend to "herd" toward the same errors (Eisenstein et al., 2024).

In this paper, we propose **Reward Model Boosting (RMB)**, where we construct a more reliable reward signal with advanced boosting methods. First, we train a group of reward models jointly with a diversity-promoting penalty; in particular, we apply the Hilbert–Schmidt Independence Criterion (HSIC; Gretton et al., 2005; 2007), which reduces the correlation among models and promotes

---

[1]Our code is released at `https://anonymous.4open.science/r/RMB-8F2C`

complementary error profiles. Then, we train a lightweight aggregator, in the principle of boosting, to combine the reward models effectively. Specifically, we learn a decision tree-based aggregator (Chen & Guestrin, 2016) that combines the reward models' outputs and iteratively minimizes the aggregated reward prediction error. The boosted decision-tree aggregator exploits reward models' complementary error profiles, yielding a more accurate and robust reward signal. Our approach offers two key advantages. *(i) Stronger aggregation.* Our RMB method learns to correct one RM's errors using the strengths of others during boosting training, thus producing more accurate predictions than simple averaging (shown in §3.4). *(ii) Robustness to text perturbation.* By using a boosted decision-tree aggregator, RMB induces a discretized mapping from model outputs to final scores, effectively smoothing spurious fluctuations caused by training noise (shown in §3.5).

We conduct extensive experiments to validate the effectiveness of our approach. First, we demonstrate that RMB significantly improves reward prediction accuracy on both in-distribution and out-of-distribution (OOD) datasets, indicating strong generalization ability (§3.2). Second, we conduct Best-of-$N$ (BoN) sampling and PPO training experiments, showing that our RMB effectively mitigates the reward hacking issue (§3.3). Third, our in-depth analysis verifies that the HSIC diversity-prompting regularizer diversifies the reward models, and that it is essential in our reward model boosting framework (§3.4). In addition, we perform text perturbation as a reward model attack, which further demonstrates the robustness of our RMB (§3.5).

## 2 METHOD

In this section, we present our RMB in detail. We first present the task formulation in §2.1, and then introduce our boosting method in §2.2. Finally, we discuss diverse reward modeling in §2.3, which is important for performance gain.

### 2.1 TASK FORMULATION

Let a dataset $D$ contain tuples of $(x, y_w, y_l)$, where $x$ is an input text, and $y_w$ and $y_l$ are two responses to $x$. A human has indicated a preference of $y_w$ over $y_l$ ($w$ for win and $l$ for loss).

Our RBM involves a set of reward models $r_1, r_2, ..., r_N$, where each model $r_i$ maps an input–response pair $(x, y)$ to a scalar reward score. The reward scores form a vector:

$$\mathbf{R}(x, y) = [r_1(x, y), r_2(x, y), \cdots, r_N(x, y)]. \tag{1}$$

Typically, a reward model is trained to assign a higher score to a preferred response $y_w$ than to a disfavored one $y_l$ by minimizing a pairwise comparison logistic loss (Ouyang et al., 2022; Gao et al., 2023; Coste et al., 2024; Yang et al., 2024; Liu et al., 2025). We further incorporate a diversity penalty, deferred to §2.3.

Our boosting objective is to learn an aggregator function $F$ combining this reward vector into a scalar reward score, which is used to evaluate the quality of a response $y$ given its input $x$ for RLHF.

### 2.2 BOOSTING THE REWARD MODELS

We propose to learn the aggregator function $F$ by adapting the gradient boosting framework, specifically following the principle of XGBoost (Chen & Guestrin, 2016). The key idea is to construct $F_K$ as an ensemble of $K$ regression decision trees $(f_1, f_2, \cdots, f_K)$, where each tree is trained on the residuals (the errors) from the combined predictions of all preceding trees. This sequential, additive modeling allows the aggregator to capture sophisticated, nonlinear interactions among the base reward model outputs.

Formally, the aggregator at iteration $k$ is expressed as:

$$F_k(\mathbf{R}(x, y)) = \sum_{t=1}^{k} f_t(\mathbf{R}(x, y)), \tag{2}$$

where each $f_t$ is a regression tree that maps the reward vector $\mathbf{R}(x, y)$ to a scalar value. This is accomplished by traversing from the root to a leaf node, guided by internal nodes that contain a split

condition based on a feature (an element in $\mathbf{R}(x, y)$) and a threshold (e.g., if $\mathbf{R}(x, y)[1] > 0.5$, go left; otherwise, go right). The final prediction is a scalar score held by a leaf node at the end of the path. The learnable parameters of a regression tree are its internal node split conditions and the scores held by the leaf nodes.

**Training objective.** Given a dataset $\mathcal{D}$ of pairwise preference annotations, where each training example is a tuple $(x, y_w, y_l)$, we aim to learn a reward function, modeled by $F_K$, that assigns a higher score to $(x, y_w)$ than to $(x, y_l)$. Following boosting, trees are added iteratively so that at iteration $k$ we fit $f_k$ to reduce the residual errors left by $F_{k-1}$. The training objective at iteration $k$ is

$$
\begin{aligned}
\mathcal{L}^{(k)} &= \sum_{(x, y_w, y_l) \in \mathcal{D}} L(F_k(\mathbf{R}(x, y_w)) - F_k(\mathbf{R}(x, y_l))) + \Omega(f_k) \\
&= \sum_{(x, y_w, y_l) \in \mathcal{D}} L\left( \underbrace{F_{k-1}(\mathbf{R}(x, y_w)) - F_{k-1}(\mathbf{R}(x, y_l))}_{\text{residual errors}} + f_k(\mathbf{R}(x, y_w)) - f_k(\mathbf{R}(x, y_l)) \right) + \Omega(f_k).
\end{aligned}
\tag{3}
$$

The data-fitting term uses the pairwise logistic loss:

$$
L(F_k(\mathbf{R}(x, y_w)) - F_k(\mathbf{R}(x, y_l))) = -\log(\sigma(F_k(\mathbf{R}(x, y_w)) - F_k(\mathbf{R}(x, y_l)))),
\tag{4}
$$

where $\sigma(z) = \frac{1}{1+e^{-z}}$. This encourages the preferred response $y_w$ to receive a higher score than $y_l$. The regularizer $\Omega(f_k) = \gamma T_k + \frac{\lambda}{2} \sum_{j=1}^{T_k} w_{kj}^2$ penalizes overly complicated trees and large leaf values, where $T_k$ is the number of leaves in tree $f_k$, and $w_{kj}$ is the value of leaf $j$, tree $k$. The coefficients $\gamma$ and $\lambda$ control the strength of the penalty.

**Learning the decision trees.** The structure is learned through a greedy algorithm, starting from a single root node and iteratively adding branches. At each node, the algorithm searches for an optimal split condition (in the form of feature thresholding). The quality of a potential split is measured by the reduction of the objective function in Eqn. (4), referred to as a gain:

$$
\text{Gain} = \mathcal{L}^{(k)}_{\text{before split}} - \mathcal{L}^{(k)}_{\text{after split}}.
\tag{5}
$$

To minimize the gain in a tractable way, we follow Chen & Guestrin (2016) and use a second-order Taylor expansion to approximate the objective function (4). Details are provided in Appendix A.1.

The process of adding tree branches stops when the best candidate gain falls below a threshold; once the tree structure is fixed, each leaf value is set by a closed form that minimizes the second-order approximation of Eqn. (4) with the $\ell_2$ penalty in $\Omega(\cdot)$. Likewise, we stop adding trees for boosting when the objective no longer improves with an additional tree.

Overall, our RMB approach iteratively minimizes the objective by discovering new tree structures, each focusing on data samples that are currently misranked. In this way, it progressively refine $F_M$ into an accurate reward model aggregator.

## 2.3 LEARNING DIVERSE REWARD MODELS FOR BOOSTING

Boosting works best when base reward models make accurate and complementary predictions. When these reward models' scores are highly correlated, they cannot contribute to loss reduction during the boosting process (Geng et al., 2007; Pan et al., 2009; Toloşi & Lengauer, 2011; Yeh & Tsai, 2022; Lyzhin et al., 2023).

A naïve approach to encourage diversity of reward model learning is to introduce stochasticity (e.g., different random seeds) in training (Coste et al., 2024; Eisenstein et al., 2024). However, models trained on the same data tend to converge to similar functions even with different random seeds (Kornblith et al., 2019; Chizat et al., 2019). We therefore explicitly introduce a diversity-promoting regularization term, in addition to the conventional reward model training objective.

**Reward modeling learning.** We follow Bradley–Terry (1952) to learn a reward model to distinguish between a preferred response $y_w$ and a less preferred one $y_l$:

$$
\mathcal{L}_{\text{reward}}(\theta) = -\mathbb{E}_{(x, y_w, y_l) \sim D} [\log(\sigma(r_\theta(x, y_w) - r_\theta(x, y_l)))],
\tag{6}
$$

where $r_\theta(x, y)$ represents the reward score for input $x$ and its corresponding response $y$ with model parameters $\theta$, and $\sigma(\cdot)$ is the sigmoid function.

**HSIC diversity regularization.** We adopt the Hilbert–Schmidt Independence Criterion (HSIC; Gretton et al., 2005; 2007) to quantify and penalize the statistical dependence between reward models. However, we do not apply HSIC on the raw reward score $r_\theta(x, y)$, as the diversity regularization may interfere the primary objective in Eqn. (6). Instead, we propose to regularize the diversity of preference margins during the training of reward models.

Let $r_{\theta_1}, \cdots, r_{\theta_N}$ be $N$ reward models and $\{(x^{(i)}, y_w^{(i)}, y_l^{(i)})\}_{i=1}^b$ be a batch of samples. We introduce a margin vector $\mathbf{m}_k$ for the $k$-th reward model:

$$\mathbf{m}_k = \begin{bmatrix} m_k^{(1)}, \cdots, m_k^{(b)} \end{bmatrix}^\top \quad \text{with} \quad m_k^{(i)} = r_{\theta_k}(x^{(i)}, y_w^{(i)}) - r_{\theta_k}(x^{(i)}, y_l^{(i)}). \tag{7}$$

These margins quantify how strongly model $k$ prefers $y_w^{(i)}$ over $y_l^{(i)}$ for input $x^{(i)}$. If two models yield nearly identical margin vectors on a batch, they are learning essentially the same notion of preference strength, i.e., collapsing to a shared margin distribution. Promoting diversity based on the margin vectors helps to learn diverse reward models in a way that they agree on sign, but differ in agreement strength.

Specifically, HSIC uses a kernel function to create a unique representation for each reward model in a high-dimensional space (Gretton et al., 2005; Ullah et al., 2024; Birjais et al., 2024). HSIC checks whether the pattern of one reward model aligns with another in the high-dimensional space based on pairs of margin vectors, given by $\text{HSIC}_b(\mathbf{m}_i, \mathbf{m}_j)$. Details of this HSIC regularizer computation are provided in Appendix A.2.

In summary, we train all $N$ reward models jointly with the following objective:

$$\mathcal{L}_{\text{total}}(\theta_1, \cdots, \theta_N) = \sum_{k=1}^N \mathcal{L}_{\text{reward}}(\theta_k) + \lambda_{\text{HSIC}} \sum_{1 \leq i < j \leq N} \text{HSIC}_b(\mathbf{m}_i, \mathbf{m}_j), \tag{8}$$

where $\lambda_{\text{HSIC}} > 0$ controls the strength of the diversity regularization. Minimizing this joint objective (8) yields an ensemble of reward models that are not only individually accurate but also exhibit diversity in the strength of preference prediction.

## 3 EXPERIMENTS

In this section, we present the empirical evaluation of our proposed RMB. We begin by describing the datasets, models, and competing approaches, followed by our main results and in-depth analyses.

### 3.1 EXPERIMENTAL SETTINGS

**Datasets.** We evaluate our approach on four human preference datasets commonly used in prior work (Zhang et al., 2025; Mandal et al., 2025b). **Unified-Feedback**[2]: One of the largest publicly available collections of pairwise human feedback. **HHH-Alignment** (Askell et al., 2021): A preference dataset constructed to evaluate language models on helpfulness, honesty, and harmlessness. **MT-Bench Human Judgments** (Zheng et al., 2024): A collection of human preferences for model responses to questions from the MT-Bench benchmark. **RewardBench** (Lambert et al., 2024): A benchmark created to evaluate the capabilities and safety alignment of reward models.

We use the Unified-Feedback dataset for training and in-distribution evaluation; we treat HHH-Alignment, MT-Bench, and RewardBench as out-of-distribution (OOD) evaluation sets, as their input and response distributions differ significantly from those of the training data.

**Models.** Following the setup of Yang et al. (2024), we use Gemma-2B-IT (Gemma et al., 2024) as the backbone for both reward model training and policy optimization; we use a publicly available Mistral-7B-Instruct-v0.2 model[3] that was fine-tuned on the Unified-Feedback dataset as the gold reward model serving for a synthetic setup to detect the reward hacking issue.

---

[2]https://huggingface.co/datasets/llm-blender/Unified-Feedback
[3]https://huggingface.co/Ray2333/reward-model-Mistral-7B-instruct-Unified-Feedback

| Approach | Unified-Feedback | HHH-Align | MT-Bench | RewardBench | | | | |
| --- | --- | --- | --- | --- | --- | --- | --- | --- |
| | | | | Avg. | Chat | Chat-Hard | Safety | Reasoning |
| Frozen RM | 63.9 | 68.6 | 68.2 | x | x | x | x | x |
| Baseline RM | 68.8 | 70.3 | 69.1 | 64.5 | 95.8 | 37.3 | 59.9 | 64.8 |
| Label Smooth | 68.5 | 68.8 | 71.9 | 61.1 | 91.6 | 39.0 | 53.8 | 60.2 |
| Margin Loss | 69.6 | 69.8 | 71.0 | 66.4 | 97.2 | 37.5 | 56.8 | **72.7** |
| GRM | 71.5 | 78.7 | 73.0 | 66.8 | 94.1 | 41.9 | 69.5 | 61.5 |
| Avg Ensemble ($N = 3$) | 69.9 | 72.2 | 71.1 | 65.2 | 96.1 | 38.2 | 58.8 | 67.6 |
| RMB ($N = 3$) | 73.64 | 81.25 | **76.34** | 68.16 | 96.35 | **44.14** | 72.40 | 65.95 |
| RMB ($N = 5$) | 74.58 | 83.20 | 76.04 | 71.65 | 95.31 | 43.36 | 78.26 | 71.42 |
| RMB ($N = 8$) | 74.55 | 84.39 | 76.23 | **72.30** | **96.61** | 43.75 | **78.39** | 72.33 |
| RMB ($N = 10$) | **75.45** | **85.07** | 75.61 | 71.00 | 96.35 | 43.96 | 77.83 | 70.42 |

Table 1: Preference prediction performance on all approaches learned using 40K data mode of Unified-Feedback. Results of competing approaches are directly sourced from Yang et al. (2024). $N$ means the number of reward models used for ensembling or boosting.

| Approach | Opponent | Pairwise Preference Comparison (in %) | | | | Pointwise Preference Comparison (in %) | | | |
| --- | --- | --- | --- | --- | --- | --- | --- | --- | --- |
| | | Win↑ | Tie | Lose↓ | Δ | Win↑ | Tie | Lose↓ | Δ |
| **RMB** ($N = 3$) | Baseline RM | 55.97 | 3.21 | 40.82 | ↑**15.15** | 44.73 | 23.97 | 31.29 | ↑**13.44** |
| | Avg Ensemble ($N = 3$) | 57.87 | 4.31 | 37.81 | ↑**20.06** | 40.22 | 34.50 | 25.28 | ↑**14.94** |
| | GRM | 54.76 | 3.01 | 42.22 | ↑**12.54** | 43.33 | 30.39 | 26.28 | ↑**7.05** |

Table 2: LLM-as-judge of policies learned via RLHF.

**Competing approaches.** We compare the performance of RMB against six competing approaches **Baseline Reward Model:** It is a standard model trained with the classic reward objective defined in Eqn. (6). **Frozen Reward Model:** Here, the reward model reuses the base language model's weights, except that a two-layer neural classifier is fine-tuned. **Margin Loss:** An additional margin is incorporated into the reward loss, as shown in Eqn. (30) (Touvron et al., 2023; Wang et al., 2024a). **Label Smoothing:** A regularization technique that penalizes overconfident model outputs to mitigate overfitting (Wang et al., 2024a). **GRM:** The GRM method employs a regularization term of the text generation objective during reward model learning (Yang et al., 2024). **Average Ensemble:** A method that averages the outputs of a group of reward models learned in the same way as the baseline reward model with different random seeds as the final reward (Coste et al., 2024). Unless stated otherwise, we average outputs of three reward models as an ensemble.

For implementation details of all approaches, please refer to the Appendix B.

## 3.2 Reward Model Performance

We evaluate the effectiveness of our approach through two main experiments: (i) reward model performance on human-preference data (i.e., whether the learned reward model can rank human preferred responses with a higher score or not), (ii) generation performance of the policy learned by RLHF based on different reward models (Ouyang et al., 2022).

**Preference prediction performance.** Following the setup of Yang et al. (2024), we learn reward models using Unified-Feedback with two training sizes: 40K samples (Tab. 1) and 400K (Tab. 7). We compute the accuracy that a reward model correctly ranks the win/lose responses (i.e., assigning a higher reward to the win response). We use the data processing code released by Yang et al. (2024) to ensure a fair comparison.

As seen from the tables, recent methods of learning reward models (such as Label Smooth (Wang et al., 2024a), Margin (Touvron et al., 2023), GRM (Yang et al., 2024), Avg Ensemble (Coste et al., 2024)) bring marginal improvement compared with the baseline approach (Ouyang et al., 2022). By contrast, our RMB approach consistently and substantially outperforms all competing methods in both in-distribution and OOD setups. In particular, our model surpasses the classic average ensemble (Coste et al., 2024) when the number of ensemble individuals is controlled ($N = 3$ in our experiments). Moreover, the performance of our RMB continues to increase with a larger $N$, until it saturates when $N$ is 8–10.

**LLM-as-judge evaluation of policies learned via RLHF.** We use an LLM judge to evaluate the policies (i.e., language models) learned by RLHF. Specifically, we use Gemma-2B-1T as a base policy model and fine-tune it by the PPO algorithm using the reward models learned in the previous

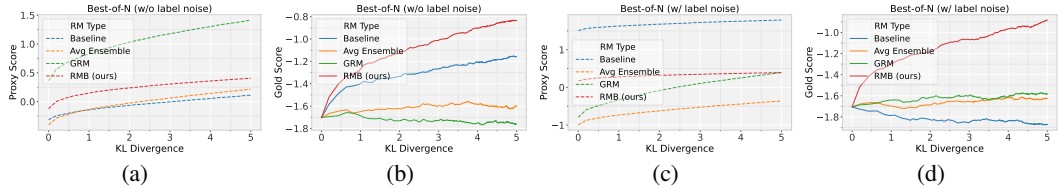

Figure 1: Results of Best-of-N sampling experiments, where both Avg Ensemble and our RMB use three individual reward models. Figures 1a and 1c intend to show the change of the proxy reward score, as an increasing proxy reward with a decreasing/plateaued gold reward indicates reward hacking. Notice that the scores of different proxy reward models are not directly comparable. Figures 1b and 1d show the gold reward scores achieved, a higher score indicating a better reward model.

setup. Then, we prompt DeepSeek-R1 (DeepSeek-AI et al., 2025) to judge the overall quality of the responses for the inputs in the Unified-Feedback hold-out test set. We select three competitors in this evaluation due to the limitation of budget: Baseline RM as a baseline method, and GRM and Avg Ensemble as strong competitors. To mitigate ID and positional biases (Zheng et al., 2023a; Shen et al., 2023), we conduct pairwise comparison in a way that each pair is queried four times by swapping the candidate order and their IDs (i.e., "A" and "B"). In addition, we also conduct a point-wise comparison, where we obtain an evaluation score ranging from 1 to 10; then, we compute a win/tie/loss rate according to the score comparison. LLM evaluation prompt templates are provided in Appendix G.

Tab. 2 reports the results of LLM-as-judge. As seen, RMB yields an LM policy that attains the highest win rate. This is consistent against all competitors and in both pairwise and pointwise settings. These findings, along with the results in the preference prediction experiment, convincingly demonstrate the effectiveness of our RMB method for RLHF.

### 3.3 EVALUATION OF THE MITIGATION OF REWARD HACKING

We analyze the reward hacking by Best-of-N (BoN) sampling and Proximal Policy Optimization (PPO) experiments, following prior work (Gao et al., 2023). This involves a synthetic data setup, where we use Mistral-7B-Instruct-v0.2 as the gold reward to provide preference annotations for training a reward model (called a proxy reward).

**BoN.** BoN sampling draws $n$ candidate responses from a general language model (i.e. Gemma-2B-1T) and picks the one with the highest score under a proxy reward model. As $n$ increases, if the chosen output's proxy score rises while its gold reward score plateaus or even declines, then it shows that this proxy reward model suffers from reward hacking. BoN sampling has been widely adopted as an experiment setup to illustrate the overoptimization issue (Gao et al., 2023; Yang et al., 2024; Liu et al., 2025).

Our experiment adopts the setup in Yang et al. (2024). We first draw a subset containing 20K samples from Unified-Feedback and annotate preference labels (i.e., which sample is preferred) based on the gold reward to train a proxy reward model. For each input in the test set, we perform BoN sampling (i.e., drawing $n$ candidates and ranking them with the proxy model). The top-ranked response is selected and then scored by the gold reward. We report both the average proxy score and the average gold score across the test inputs. We vary the selection budget via the BoN KL divergence from 0 to 5, corresponding to the number of responses $n$ ranging from 1 to 405 per input. The KD divergence is given by $\mathrm{KL}_{\mathrm{BoN}} = \log n - \frac{n-1}{n}$, which quantifies how $n$-category uniform distribution deviates from the one-hot distribution of a sample.

Figs 1a and 1b show the performance of our RMB approach and competing approaches in the BoN sampling experiment. We see that, for competing approaches, the gold reward score increases much less than the proxy reward score as KL grows, indicating reward hacking exists. By contrast, RMB exhibits a clear, monotonic increase in the gold reward score with larger KL, demonstrating its advantage in mitigating reward hacking.

It is noticed that human-preference data typically contains $\sim$ 20–30% label noise (Wang et al., 2024a), which may degrade reward model generalization (Ramé et al., 2024; Liang et al., 2024) and hinder policy learning (Ye et al., 2024; Mandal et al., 2025a). To assess robustness under label noise, we randomly choose 25% of the data samples and flip their gold reward-induced preference labels.

The results in Figs 1c and 1d show that, with label noise, the baseline suffers from significant reward hacking, as its gold reward score decreases. GRM and Avg Ensemble demonstrate some

Figure 2: PPO training with RMs. Ensemble and RMB both adopt three RMs to aggregate.

| | | **Unified-Feedback** (In-Distribution) | | | | | **RewardBench** (Out-of-Distribution) | | | | |
|---|---|---|---|---|---|---|---|---|---|---|---|
| Random Seeds Diversity | Individual RMs | 70.71 | 71.34 | 70.52 | 69.85 | 70.30 | 68.45 | 67.66 | 69.97 | 67.11 | 67.24 |
| | Avg Ensemble | | | 70.80 | | | | | 68.92 | | |
| | Boosting | | | 71.04 | | | | | 68.70 | | |
| HSIC Diversity | Individual RMs | 73.40 | 70.73 | 69.29 | 68.05 | 73.84 | 70.08 | 69.95 | 67.58 | 66.97 | 71.62 |
| | Avg Ensemble | | | 72.04 | | | | | 69.40 | | |
| | Boosting | | | 74.58 | | | | | 71.65 | | |

Table 3: Avg Ensemble vs Boosting with different diversity strategy. We also report the performance of individual RMs (index from 0 to 4).

capacity of mitigating reward hacking, but the gold reward score achieved still plateaus as their proxy reward scores continue to increase. By contrast, RMB achieves the highest gold reward score, which increases robustly. This indicates that our approach can reliably estimate the response quality even when trained on noisy preference labels.

**PPO.** PPO (Schulman et al., 2017) is a classic RL algorithm designed to learn a policy maximizing expected rewards. It has been widely adopted in RLHF to learn a language model policy maximizing the preference score provided by reward models (Ouyang et al., 2022; Achiam et al., 2023; Liu et al., 2024) and also serves as a standard testbed for studying the reward hacking issue (Gao et al., 2023; Moskovitz et al., 2024; Bukharin et al., 2025).

Specifically, we adopt the Gemma-2B-1T model as an initial policy, and use the PPO algorithm to optimize this policy based on the proxy reward model (which is learned by gold reward-induced preference labels). The PPO-finetuned policy is then used to generate responses on the inputs from the test set of Unified-Feedback. We track the changes of both proxy and gold reward scores of the generated response throughout PPO training to assess whether improvement under the proxy reward score translates to that under the gold reward score.

As seen from Figs2a and 2b, all methods increase the proxy reward score at the beginning of training; however, competing approaches exhibit noisier trajectories with a noticeable mid-training drop, whereas RMB shows smoother and more stable improvements. For the gold score, competing approaches improve briefly only in the first few steps and collapse later. On the contrary, the curve of the gold reward in our approach is more reflected by the proxy reward curve. The results suggest that our approach is able to substantially mitigate reward hacking.

As in the BoN setting, we also have a setup with $25\%$ noise added to preference labels in the PPO experiment, and the learning curves are shown in Figs2c and 2d. As expected, the gold reward score is more fluctuated due to the added label noise. But still, our RMB maintains the highest and least decreasing gold score, demonstrating that our approach is more robust than competing methods with noisy preference labels.

### 3.4 ANALYSIS OF THE COMPONENTS OF OUR RMB

We perform an in-depth analysis of two main components of our approach, namely, the boosting method and the HSIC diversity-promoting regularization. We follow most of the setups in §3.2, except that we consider five[4] individual reward models (RMs) by either varying random seeds or using our HSIC diversity-promoting regularizer.

As shown in Tab. 3, Average Ensemble and our RMB achieve similar performance in the random seed setup, whereas with HSIC-induced diversity, our boosting considerably outperforms Average Ensemble. This shows that different random seeds yield similar reward models, thus making ensem-

---

[4]In the main experiment, we try different numbers of individual ($N = 3, 5, 8, 10$) for our RMB, but compare our approach with Avg Ensemble in the setting of $N = 3$ as we quote results from previous work for Tab. 1. Here, we consider five individual RMs to have a more reliable correlation analysis.

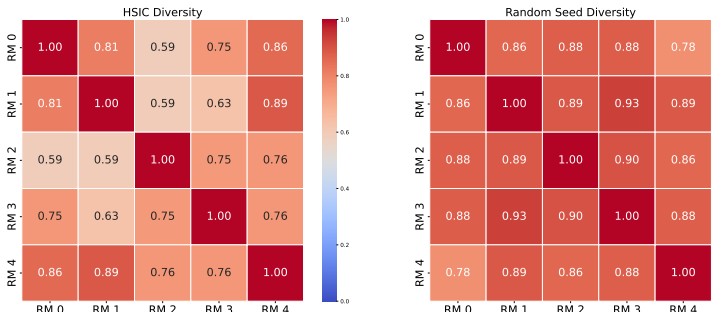

Figure 3: Correlation heat map among individual Reward Models

bles less effective. By contrast, our HSIC encourages complementary error profiles that a boosting aggregator can leverage to achieve substantially higher performance.

The correlation among RMs can be further viewed from the heatmaps in Fig. 3. For each pair of RMs, we measure the ratio of pairwise-preference agreement on the Unified-Feedback test set. RMs learned under random seed diversity exhibit very high pairwise agreement (most pairs $> 80\%$), indicating strong correlations; HSIC-trained RMs are less correlated (typically 50–80%), providing the diversity that boosting needs.

We further examine the contribution of each RM during the boosting process. This is measured by *Total Gain* (Chen & Guestrin, 2016), i.e., the sum of gains over all splits (see Eqn. (5)) that each RM obtains in the decision trees in Tab. 4. We see that, under the random seeds setup, Total Gain is heavily skewed (e.g., a single RM contributes 70.13%). This is because such RMs tend to be redundant and thus are ineffective when the boosting process minimizes residual errors.

| HSIC Diversity | Total Gain | Random Seeds Diversity | Total Gain |
|---|---|---|---|
| RM 0 | 29.75% | RM 1 | 70.13% |
| RM 4 | 27.39% | RM 0 | 12.39% |
| RM 1 | 26.78% | RM 4 | 10.99% |
| RM 2 | 8.55% | RM 2 | 5.82% |
| RM 3 | 7.53% | RM 3 | 0.66% |

Table 4: Total Gain ratio of individual RMs.

Under the HSIC setup, the Total Gain is distributed more evenly. Each of RM 0, RM 4, and RM 1 contributes 25–30%, and they are the strong individual RMs (seen in Tab. 3). This shows that boosting emphasizes the best RMs while using the remaining RMs to correct residual errors. Overall, our boosting approach surpasses every individual RM.

**Discussion.** The low correlation of individuals is shown to be important for boosting algorithms in commercial ranking systems, such as search and recommendation (Covington et al., 2016; Wang et al., 2022; Xi et al., 2023), for which researchers perform heavy feature engineering and propose heterogeneous signals (e.g., semantic relevance, like count, and comment count) to be considered during boosting (Ni et al., 2021; Haldar et al., 2023; Du et al., 2024). In our setting, HSIC serves as an auxiliary objective that encourages low pairwise correlation across RMs, which is analogous to the feature engineering process of traditional boosting systems.

### 3.5 ANALYSIS OF ROBUSTNESS UNDER ATTACKS

We evaluate RM robustness under small, meaning-preserving text perturbations. This is accomplished by comparing how an RM ranks a set of response candidates for the same input, with and without perturbation. Specifically, we sample 400 candidate responses for each input in the test set of Unified-Feedback using a pretrained Gemma-2B-1T. We first compute reward scores on the original texts and obtain a ranking in descending order for each input. We then apply a keyboard-neighbor typo attack to the text by randomly replacing characters with adjacent keys on the QWERTY layout (Belinkov & Bisk, 2018; Pruthi et al., 2019), and compute a new rank. To study sensitivity, we vary the perturbation budget (number of character edits per example) in $\{1, 3, 5, 8, 10\}$.

We measure the similarity of the two rankings by three metrics: (1) **Pairwise accuracy**: the ratio of correctly ordered pairs among all pairs, (2) **Footrule similarity**: one minus normalized Spearman's footrule distance, which sums the absolute differences between the ranks of each item in the two rankings (Diaconis & Graham, 1977), and (3) **Normalized Discounted Cumulative Gain (nDCG)**: a metric that measures the usefulness of items based on their position in the rank (Järvelin & Kekäläinen, 2002), where we treat the original order as the ground truth. All metrics are in the range of $[0, 1]$, and the higher, the better.

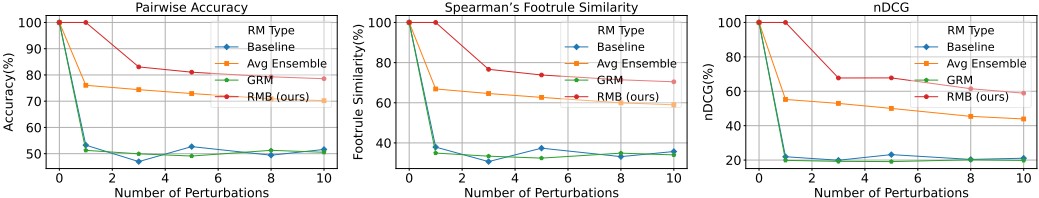

Figure 4: Rank Order Preservability.

Figure 4 shows the results of our RMB ($N = 3$), along with the Baseline RM, Avg Ensemble ($N = 3$), and GRM. As seen, our RMB attains the highest scores on all three metrics, indicating that our RMB is more robust than competing approaches under small, meaning-preserving attacks.

**Discussion.** Decision-tree-based aggregators are known to be robust in ranking tasks (Joachims et al., 2017; Wang et al., 2018; Haldar et al., 2019; Wu et al., 2022), which partially explains that our RMB can maintain high performance during attacks. It is interesting to notice that a Transformer (Vaswani et al., 2017)-based reward model maps discrete text to continuous reward values, and a small change of the input text may lead to large changes of the output, especially when the training set is small (Wu et al., 2025); such a phenomenon is observed in other tasks as well (Ebrahimi et al., 2018; Jin et al., 2020). Our RMB performs aggregation of individual RMs by only predicting one of its leaf values, which are a finite discrete set. This reduces spurious fluctuations caused by training noise, making our model more robust under attacks.

### 3.6 HSIC RATIO ANALYSIS

The selection of the diversity regularization coefficient, $\lambda_{HSIC}$, follows the established practice of balancing the main task loss with the diversity penalty (Dong et al., 2023; Ma et al., 2024). In this appendix, we perform a sensitivity analysis by sweeping $\lambda_{HSIC}$ across four orders of magnitude (from 0.001 to 10) to evaluate its impact on model performance. We report results on the Unified-Feedback dataset, utilizing the same experimental setup detailed in Section 3.4.

| $\lambda_{\text{HSIC}}$ | Avg. Indiv. RM Perf. | Avg. Pairwise Corr. | RMB Perf. |
|---|---|---|---|
| 0.001 | $70.7 \pm 0.8$ | 92% | 71.7 |
| 0.01 | $71.3 \pm 1.1$ | 86% | 72.8 |
| 0.1 | $71.1 \pm 2.5$ | 73% | **74.6** |
| 1 | $68.8 \pm 3.0$ | 59% | 71.2 |
| 10 | $61.7 \pm 3.7$ | 51% | 66.5 |

Table 5: Sensitivity analysis of the $\lambda_{\text{HSIC}}$. We report the average performance of individual Reward Models, the average pairwise correlation between them, and the final RMB performance.

As presented in Table 5, our approach demonstrates strong robustness to the choice of $\lambda_{HSIC}$. The performance of RMB remains stable and effective across a broad range of values ($0.001 \leq \lambda_{HSIC} \leq 1$), even as the average pairwise correlation among reward models varies significantly (from 92% down to 59%). This indicates that RMB can effectively leverage increasing diversity to boost performance, provided the base estimators maintain reasonable accuracy. Performance deterioration is observed only when the regularization becomes overwhelmingly dominant ($\lambda_{HSIC} = 10$). In this regime, while diversity is maximized (correlation drops to 51%), the aggressive penalty interferes with the optimization of the main reward objective. Consequently, the accuracy of individual reward models degrades sharply (61.7%), which in turn limits the effectiveness of the boosted aggregator.

## 4 RELATED WORK

The challenge of reward hacking in RLHF was empirically established by (Ouyang et al., 2022; Gao et al., 2023). There are generally two directions to mitigate the reward hacking issue: (i) improving the RL training process under an imperfect reward signal, and (ii) learning a better reward model to provide a reliable training signal for RL.

**Optimizing the RL process.** One line of work mitigates reward hacking by refining the RL optimization process under imperfect reward signals. For example, researchers update the policy conservatively by designing pessimistic reward signals given an imperfect RM (Zhai et al., 2023; Zhang et al., 2024; Yan et al., 2024; Mandal et al., 2025b). Another strategy is to maintain a pretrained

policy's text generation ability by regularizing the RL objective (Ouyang et al., 2022; Liu et al., 2024; Laidlaw et al., 2025), or averaging policies' parameters (Lin et al., 2024). Researchers also constrain the policy optimization by restricting the exploration region (Moskovitz et al., 2024; Dai et al., 2025; Mandal et al., 2025a).

**Improving reward models.** Our work falls into this category, which combats reward hacking by strengthening reward models. Building an ensemble of multiple RMs can stabilize noisy reward signals; for example, Coste et al. (2024) average RM outputs and Ramé et al. (2024) average RM parameters. However, simple averaging cannot remove the bias shared among individual RMs (Eisenstein et al., 2024). To address this, our RMB learns an aggregator in the principle of boosting (Chen & Guestrin, 2016), which reduces residual reward error in a complementary way.

Most other RM-centric studies focus on improving an individual reward model, including designing better RM training objectives (Yang et al., 2024; Wang et al., 2024b), optimizing RM architectures (Miao et al., 2024; Chen et al., 2024), and improving training data quality (Zhu et al., 2024; Wolf et al., 2025; Haider et al., 2025; Liu et al., 2025; Bukharin et al., 2025). These approaches focus on a single RM, which can be potentially combined with our boosting approach for further performance improvement.

## 5 CONCLUSION

To address the challenge of reward hacking in RLHF, we propose Reward Model Boosting (RMB). Our method learns a decision-tree aggregator over an ensemble of reward models to produce a more accurate and robust reward signal. We further adopt the HSIC diversity-promoting regularizer for reward model training. Our experimental results show that RMB enhances preference prediction accuracy across both in-distribution and out-of-distribution datasets. Moreover, our RMB effectively mitigates reward hacking and stays robust when countering text perturbation. Overall, RMB provides a reliable reward signal for training policies to align human preferences.

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

# A RMB Details

## A.1 Optimization of the Reward Model Boosting Objective

This section provides a detailed derivation for learning the tree aggregator, $f_m$, at each boosting iteration $m$. The primary goal is to determine the optimal tree structure and leaf values that minimize the overall objective function defined in Eqn. (4). The optimization follows the principle of XGBoost (Chen & Guestrin, 2016).

**Objective function approximation.** The training objective at the $m$-th iteration, as presented in Eqn. (4) of the main text, can be rewritten as the following form:

$$\mathcal{L}^{(k)} = \sum_{(x,y_w,y_l)\in\mathcal{D}} L(\Delta F_{k-1}(\mathbf{R}) + \Delta f_k(\mathbf{R})) + \Omega(f_k), \qquad (9)$$

where $\Delta F_{k-1}(\mathbf{R}) = F_{k-1}(\mathbf{R}(x,y_w)) - F_{k-1}(\mathbf{R}(x,y_l))$ and $\Delta f_k(\mathbf{R}) = f_k(\mathbf{R}(x,y_w)) - f_k(\mathbf{R}(x,y_l))$. $\Omega(f_k)$ is a regularizer of a tree to penalize overly complicated trees structure and large leaf values To facilitate efficient optimization, we approximate the loss function $L$ using a second-order Taylor expansion around the prediction from the previous iteration, $\hat{y}^{(k-1)} = \Delta F_{k-1}(\mathbf{R})$, converting the problem into minimizing a quadratic form.

Let $g_m$ and $h_m$ be the first and second-order gradients of the loss function with respect to the prediction for the $m$-th preference data point in the training dataset, denoted by

$$g_m = \left.\frac{\partial L(y_m, \hat{y})}{\partial \hat{y}}\right|_{\hat{y}=\hat{y}_m^{(k-1)}} \qquad (10)$$

$$h_m = \left.\frac{\partial^2 L(y_m, \hat{y})}{\partial \hat{y}^2}\right|_{\hat{y}=\hat{y}_m^{(k-1)}} \qquad (11)$$

The loss function for a single instance can thus be approximated as:

$$L(\hat{y}_m^{(k-1)} + \Delta f_k(\mathbf{R})) \approx L(\hat{y}_m^{(k-1)}) + g_m \Delta f_k(\mathbf{R}) + \frac{1}{2} h_m \Delta f_k^2(\mathbf{R}). \tag{12}$$

After removing the constant term $L(\hat{y}_m^{(k-1)})$, we simplify the objective function to:

$$\tilde{\mathcal{L}}^{(k)} \approx \sum_{m=1}^{M} \left[ g_m \Delta f_k(\mathbf{R}_m) + \frac{1}{2} h_m \Delta f_k^2(\mathbf{R}_m) \right] + \Omega(f_k). \tag{13}$$

**Expressing the objective by the parameters of decision trees.** A decision tree $f_k$ is defined by its structure $q$ and leaf values $\mathbf{w}$. The structure $q : \mathbb{R}^d \rightarrow \{1, \cdots, T\}$ maps an input instance $\mathbf{R} \in \mathbb{R}^N$ to a specific leaf index in $1, \cdots, T$, and $w_j$ is the score assigned to the $j$th leaf node for $j = 1, \cdots, T$.

To express the parameters explicitly in the objective in Eqn. 13, we regroup the sum over training instances to a sum over the disjoint sets of instances contained in each leaf. In particular, the term $\Delta f_k(\mathbf{R})$ can be expressed as the difference between the leaf values corresponding to the winning and losing responses: $\Delta f_k(\mathbf{R}) = w_{q(\mathbf{R}_w)} - w_{q(\mathbf{R}_l)}$. Let $I_j = \{k : q(\mathbf{R}_k) = j\}$ denote the set of instances partitioned into leaf $j$. By regrouping the summation in Eqn. (13) over the leaves, we obtain:

$$\tilde{\mathcal{L}}^{(k)} = \sum_{j=1}^{T} \left[ \left( \sum_{m \in I_j} g_m \right) w_j + \frac{1}{2} \left( \sum_{m \in I_j} h_m \right) w_j^2 \right] + \gamma T + \frac{1}{2} \lambda \sum_{j=1}^{T} w_j^2 \tag{14}$$

where the last two terms are the regularizer, i.e., $\Omega(f_k) = \gamma T + \frac{1}{2} \lambda \sum_{j=1}^{T} w_j^2$. Let us define the sum of gradient statistics in each leaf as $G_j = \sum_{m \in I_j} g_m$ and $H_j = \sum_{m \in I_j} h_m$. The objective can be concisely written as:

$$\tilde{\mathcal{L}}^{(k)} = \sum_{j=1}^{T} \left[ G_j w_j + \frac{1}{2} (H_j + \lambda) w_j^2 \right] + \gamma T. \tag{15}$$

**Optimal leaf weight calculation.** The objective function in Eqn. (15) is a sum of independent quadratic functions of each leaf weight $w_j$. To find the optimal value $w_j^*$, we take the partial derivative of $\tilde{\mathcal{L}}^{(k)}$ with respect to $w_j$ and set it to zero:

$$\frac{\partial \tilde{\mathcal{L}}^{(k)}}{\partial w_j} = G_j + (H_j + \lambda) w_j = 0, \tag{16}$$

which in turn yields the optimal leaf weight:

$$w_j^* = -\frac{G_j}{H_j + \lambda}. \tag{17}$$

**Decision-tree structure learning.** Finding the globally optimal tree structure is difficult, as it would require evaluating an exponential number of possible tree structures. Instead, we adopt a greedy, top-down approach to build the tree by making the best possible split at each step, which is computationally feasible and effective in practice.

We use *gain* in Eqn. (5) to quantify the quality of a potential split. Specifically, we first substitute the optimal leaf weight $w_j^*$ in Eqn. (17) back into the objective function in Eqn. (15). This yields a score that evaluates the quality of a given tree structure $q$:

$$\tilde{\mathcal{L}}^{(m)}(q) = -\frac{1}{2} \sum_{j=1}^{T} \frac{G_j^2}{H_j + \lambda} + \gamma T. \tag{18}$$

Consider a split that partitions a parent leaf into two child leaves, left and right. The gain of this split is defined as the reduction in the objective function:

$$\text{Gain} = \tilde{\mathcal{L}}_{\text{parent}} - (\tilde{\mathcal{L}}_{\text{left\_child}} + \tilde{\mathcal{L}}_{\text{right\_child}}). \tag{19}$$

Combining Eqn. (18), we have

$$\text{Gain} = \frac{1}{2} \left[ \underbrace{\frac{G_L^2}{H_L + \lambda}}_{\text{Left Score}} + \underbrace{\frac{G_R^2}{H_R + \lambda}}_{\text{Right Score}} - \underbrace{\frac{G^2}{H + \lambda}}_{\text{Parent Score}} \right] - \underbrace{\gamma T}_{\text{Penalty}}. \tag{20}$$

The algorithm enumerates all possible splits over all entries in $\mathbf{R}$ and selects the split that maximizes this gain. The process is repeated until a stopping criterion is met (i.e., reaching a maximum depth or observing no positive gain).

## A.2 DETAILS OF THE HSIC DIVERSITY-PROMOTING REGULARIZER

In the §2.3, we aim to train a diverse ensemble of $N$ reward models $\{R_{\theta_n}\}_{n=1}^N$. Specifically, we apply the Hilbert–Schmidt Independence Criterion (HSIC; Gretton et al., 2005; 2007) to the preference margins.

Consider a batch of $b$ preference pairs $D_b = \{(x_w^{(i)}, x_l^{(i)})\}_{i=1}^b$ and two reward models $R_{\theta_m}$ and $R_{\theta_n}$. We calculate the preference margin for the $i$th sample by

$$\mathbf{p}_m^{(i)} = R_{\theta_m}(x_w^{(i)}) - R_{\theta_m}(x_l^{(i)}), \tag{21}$$

$$\mathbf{p}_n^{(i)} = R_{\theta_n}(x_w^{(i)}) - R_{\theta_n}(x_l^{(i)}). \tag{22}$$

We group the preference margins of different samples as vectors, denoted by $\mathbf{p}_m$ and $\mathbf{p}_n$. HSIC assumes that the entries in $\mathbf{p}_m$ and $\mathbf{p}_n$ come from their respective underlying distributions, and the goal is to derive an empirical estimator for the dependence between the two distributions.

**HSIC dependency measure.** While the notion of covariance is a standard measure of dependency, it is restricted to linear correlation. The HSIC provides a more general way of capturing non-linear dependency by analyzing the relationship between variables in a high-dimensional feature space through kernel methods.

The HSIC begins by projecting the random variables $p_m$ and $p_n$ from their input space $\mathcal{X}$ onto a Reproducing Kernel Hilbert Space (RKHS), denoted $\mathcal{F}$, via a feature map $\phi(\cdot) : \mathcal{X} \to \mathcal{F}$. For an inner-product of $\phi(a)$ and $\phi(b)$, a computationally tractable kernel function is used

$$\langle \phi(a), \phi(b) \rangle_\mathcal{F} = k(a, b). \tag{23}$$

Thus, we do not need to explicitly compute $\phi(\cdot)$. A common choice for the kernel is the Gaussian RBF kernel, $k(a, b) = \exp\left(-\frac{\|a-b\|^2}{2\sigma^2}\right)$, where the hyperparameter $\sigma$ controls its sensitivity.

Within this framework, the concept of covariance is generalized by the cross-covariance operator:

$$C_{mn} = \mathbb{E}_{p_m, p_n}\left[\left(\phi(p_m) - \mu_m\right) \otimes \left(\phi(p_n) - \mu_n\right)\right], \tag{24}$$

where $\mu_m = \mathbb{E}_{p_m}[\phi(p_m)]$ and $\mu_n = \mathbb{E}_{p_n}[\phi(p_n)]$ are the means. The tensor product $\otimes$ captures the complete covariance structure between the feature vectors.

HSIC is then defined as the squared Hilbert–Schmidt norm of this operator, which quantifies its magnitude:

$$\text{HSIC}(p_m, p_n) = \|C_{mn}\|_{HS}^2 \tag{25}$$

$$= \left\|\mathbb{E}_{p_m, p_n}\left[\left(\phi(p_m) - \mu_m\right) \otimes \left(\phi(p_n) - \mu_n\right)\right]\right\|_{HS}^2 \tag{26}$$

While the theoretical definition of HSIC is based on expectations over a distribution, in practice we must compute an empirical estimate from a finite set of data samples. Let $\{(p_m^{(i)}, p_n^{(i)})\}_{i=1}^b$ be $b$-many pairs of samples in a batch. The kernel matrices for the two variables are $\mathbf{K}_m \in \mathbb{R}^{b \times b}$ and $\mathbf{K}_n \in \mathbb{R}^{b \times b}$, given by

$$(\mathbf{K}_m)_{ij} = k\big(p_m^{(i)}, p_m^{(j)}\big), \qquad (\mathbf{K}_n)_{ij} = k\big(p_n^{(i)}, p_n^{(j)}\big). \tag{27}$$

Table 6: Key implementation details of our main experiments.

| Basic information | |
| --- | --- |
| Base models | Gemma-2b-it |
| Quantization for training | bf16 |
| Fine-tuning strategy | LoRA |
| LoRA $r$ | 32 |
| LoRA alpha | 64 |
| LoRA dropout | 0.05 |
| Optimizer | Adamw_hf |
| Batch size | 16 |
| Learning rate | $1 \times 10^{-5}$ |
| Learning rate scheduler | cosine |
| Warmup ratio | 0.03 |
| **RMB** | |
| HSIC ratio $\lambda$ | 0.1 |
| Max depth | 5 |
| Max round of boosting | 256 |
| L2 regularizer | 0.1 |
| **PPO** (Schulman et al., 2017) | |
| KL regulaization | 0.0 |
| Epochs | 1 |
| Learning rate | $1 \times 10^{-5}$ |
| Lambda for GAE | 0.95 |
| Gamma | 1 |
| Clip range | 0.2 |
| Number of optimization Epochs per batch | 4 |
| Number of tokens during Generation | 512 |

The empirical estimate for HSIC from this finite batch is given by a matrix formula:

$$\text{HSIC}_b(\mathbf{p}_m, \mathbf{p}_n) = \frac{1}{(b-1)^2} \, \text{tr}\big(\mathbf{K}_m \, \mathbf{H} \, \mathbf{K}_n \, \mathbf{H}\big), \tag{28}$$

where $\mathbf{H} \in \mathbb{R}^{b \times b}$ is the centering matrix, defined as

$$\mathbf{H} \;=\; \mathbf{I} - \frac{1}{b} \, \mathbf{1}\mathbf{1}^{\mathsf{T}}. \tag{29}$$

Here, $\mathbf{I}$ is the identity matrix and $\mathbf{1}$ is a column vector of ones. Overall, the HSIC serves as a regularizer in the reward model training objective in Eqn. (8), offering a computationally feasible way to capture non-linear dependency, and thus promotes diversity in our learned reward models.

## B  IMPLEMENTATION DETAILS

**Reward model architecture.**  Base reward models use the *AutoModelForSequenceClassification* architecture from Wolf et al. (2020). The trainable parameters are a reward head and LoRA (Hu et al., 2022) adapters. The reward head is two-layer feedforward network with 1024 dimensions and a ReLU activation function, which follows the setups in Yang et al. (2024). LoRA hyperparameters are listed in Tab. 6. All reward model learning approaches employ the same architectures and have the same trainable parameters.

**Details of competing methods.** We discuss the competing methods as follows.

- **Baseline.** This is a vanilla model trained to minimize the objective in Eqn. (4).
- **Training with labeled margin** (Wang et al., 2024a). This approach modifies the classic objective by enforcing that $y_w$ exceeds $y_l$ by a certain margin $m$:

$$\mathcal{L}_{\text{margin}}(\theta) = -\mathbb{E}_{(x,y_w,y_l) \sim D}[\log \sigma(r_\theta(x, y_w) - r_\theta(x, y_l) - m)]. \tag{30}$$

The Unified-Feedback dataset provides a margin label $m$ for each pairwise sample, which indicates the desired reward difference. Note that such an approach requires additional

| Reward model | Unified-Feedback | HHH-Align | MT-Bench | RewardBench | | | | |
| --- | --- | --- | --- | --- | --- | --- | --- | --- |
| | | | | Avg. | Chat | Chat-Hard | Safety | Reasoning |
| Frozen RM | 63.8 | 66.4 | 69.5 | x | x | x | x | x |
| Baseline RM | 72.1 | 73.4 | 71.2 | 68.2 | 95.5 | 38.0 | 73.8 | 65.3 |
| Label Smooth | 71.5 | 72.1 | 71.2 | 70.6 | 94.4 | 37.3 | 73.2 | **77.4** |
| Margin Loss | 72.0 | 75.0 | 72.6 | 70.2 | 95.8 | 38.4 | 73.9 | 72.5 |
| GRM | 73.2 | 79.8 | 73.4 | 70.8 | 97.8 | 42.1 | 77.9 | 65.2 |
| Avg Ensemble ($N = 3$) | 72.8 | 76.8 | 73.7 | 71.0 | **98.0** | 37.5 | 77.3 | 71.3 |
| RMB ($N = 3$) | 74.39 | 81.80 | 75.24 | 71.97 | 95.57 | **44.14** | 78.12 | 71.81 |
| RMB ($N = 5$) | 75.12 | 82.64 | 75.63 | 72.23 | 91.85 | 39.44 | **80.72** | 73.68 |
| RMB ($N = 8$) | **75.92** | **83.04** | **76.56** | **73.34** | 95.05 | 43.54 | 76.56 | 74.51 |
| RMB ($N = 10$) | 74.93 | 82.14 | 76.18 | 72.62 | 96.88 | 42.77 | 77.86 | 73.31 |

Table 7: Preference prediction performance on all Reward Models learned using 400K data mode of Unified-Feedback. Results of competing approaches are directly sourced from Yang et al. (2024). $N$ means the number of reward models used for ensembling or boosting.

labels (i.e., a margin label for every sample) compared with our method, and thus it may not be applicable to other preference datasets.

- **Label smoothing** (Wang et al., 2024a). To account for potential annotation noise, this approach assumes that a labeled pair is flipped with probability of $\epsilon$ (set to 0.1). Then the training objective thus becomes

$$\mathcal{L}_{\text{smooth}}(\theta) = -\mathbb{E}_{(x,y_w,y_l)\sim D}[(1 - \epsilon) \log \sigma(r_\theta(x, y_w) - r_\theta(x, y_l)) + \epsilon \log \sigma(r_\theta(x, y_l) - r_\theta(x, y_w))], \tag{31}$$

This improves robustness to label errors and mitigates overfitting.

- **Avg Ensemble** (Coste et al., 2024). For this competing approach, we train three reward models (in the same way as the Baseline approach) with different random seeds and aggregate their outputs by simple averaging.
- **GRM** (Yang et al., 2024). GRM regularizes reward model learning with a text generation objective:

$$\mathcal{L}_{\text{GRM}}(\theta, \theta_{\text{LM}}) = -\mathbb{E}_{(x,y_w,y_l)\sim D}[\log(\sigma(r_\theta(x, y_w) - r_\theta(x, y_l)))] - \alpha\mathbb{E}_{(x,y_c)\sim D}[\log \sigma(\beta \log(\pi_{\theta_{\text{LM}}}(y_c \mid x)))]. \tag{32}$$

where we set $\beta = 0.1$ and $\alpha = 0.0005$ following the best setting reported by Yang et al. (2024).

**Computing resources.** Experiments are run on NVIDIA RTX A6000 (48 GB) GPUs. Training a 2B-parameter reward model with LoRA (Hu et al., 2022) on 40K training examples for two epochs requires approximately 31.9 GPU–hours.

## C  ADDITIONAL RESULTS

### C.1  RESULTS WITH A DIFFERENT DATA SIZE

Following the experimental setup of Yang et al. (2024), we trained our reward models on the Unified-Feedback dataset with both 40K and 400K samples, as described in §3.2. In the main paper, we only present the 40K-sample experiment (Tab. 1) due to the limit of space. In this appendix, we show the results of the 400-sample experiment in Tab. 7. We see that our RMB model surpasses all competing methods on both in-distribution and out-of-distribution (OOD) datasets. These findings are consistent with the performance observed in the 40K-sample experiment (Tab. 1).

### C.2  EFFICIENCY ANALYSIS

Our RMB method introduces a lightweight aggregator on top of a set of base Reward Models (RMs). To verify its efficiency, we evaluate its impact on inference latency. Specifically, we decompose the

| Section | Latency | Mean | p50 | p90 | p99 |
|---------|---------|------|-----|-----|-----|
| Breakdown (s/batch) | Base RMs | 19.7915 | 19.7442 | 20.2839 | 20.3000 |
|  | Avg Ensemble aggregator | $5.92\times10^{-5}$ | $5.67\times10^{-5}$ | $6.09\times10^{-5}$ | $1.398\times10^{-4}$ |
|  | RMB aggregator | $1.7458\times10^{-3}$ | $1.6997\times10^{-3}$ | $1.9010\times10^{-3}$ | $2.5694\times10^{-3}$ |
| End-to-End (s/batch) | Avg Ensemble | 19.7915 | 19.7443 | 20.2840 | 20.3001 |
|  | RMB | 19.7932 | 19.7460 | 20.2857 | 20.3016 |

Table 8: Inference-efficiency summary.

inference process into two stages: (1) the **base RM pass**, which involves a forward pass through each of the three RMs to generate individual scores, and (2) the **aggregation step**, where individual RM scores are combined into a final prediction. We measure the time spent in each stage for both our RMB aggregator and Average Ensemble (Coste et al., 2024). Metrics include

- **Latency** (s/batch): The wall-clock time required for a given stage. We report the mean, median (p50), 90th (p90), and 99th (p99) percentile latencies across all batches.
- **End-to-end latency** (s/batch): The overall latency including RMs and the aggregator, as well as other code components (such as IO).

The results are summarized in Tab. 8. As seen, the main computation is devoted to the base RM pass, with a mean latency of $19.791\,\mathrm{s}$ per batch. In comparison, the Average Ensemble aggregator adds a mean overhead of only $5.92 \times 10^{-5}\,\mathrm{s}$ ($\approx 0.06\,\mathrm{ms}$). Similarly, our RMB aggregator is also efficient as the overhead is magnitudes lower than the base RM pass, as the aggregator–end2end ratio is only $0.000088$. The results show that our RMB is a lightweight aggregator. It introduces a negligible latency to the reward model inference. Thus, it also introduces a negligible latency to RLHF training when the rest of the algorithm and implementation are controlled.

# D  TRAINING EFFICIENCY

This appendix presents a comprehensive analysis of the computational efficiency of RMB, detailing the breakdown of training costs and examining the trade-off between performance and efficiency as the number of reward models, $N$, increases. We demonstrate that RMB incurs a training cost comparable to existing ensemble-based approaches and that the method remains robust across reasonable choices of $N$.

**Reward-Model Training Cost**   The primary computational overhead in RMB stems from training the base RMs. As the number of RMs increases, the training cost grows linearly. In our experimental setup, training a single RM per epoch of data requires approximately 31.9 GPU-hours on an NVIDIA RTX A6000 (48GB). However, the most appropriate comparison is not against a single, weak RM, but rather against other methods that employ multiple RMs to mitigate reward hacking, such as reward model ensembles (Coste et al., 2024; Eisenstein et al., 2024). When compared under a fixed budget of $N$ trained RMs, RMB proves to be both efficient and effective. For example, with $N = 3$, RMB outperforms an average ensemble using $N = 3$ or even $N = 5$ on both preference prediction and downstream RLHF tasks. This indicates that RMB achieves superior performance with lower total computational expenditure than competing ensemble-based methods, which often require training a larger number of models to achieve similar or weaker gains.

**Cost of HSIC Diversity Regularization**   RMB jointly trains $N$ RMs with a diversity-promoting regularizer based on the HSIC. This term is computed over per-batch margin vectors and is implemented via a small number of matrix operations. In practice, the computational overhead of this regularization is negligible relative to the forward and backward passes required for the $N$ large language models serving as RMs.

**Boosting Aggregator Training Cost**   Following the training of the RMs, RMB learns a tree-based boosting aggregator on the frozen RM outputs. This stage is extremely lightweight; in our experiments, training the aggregator on the full dataset requires less than 5 minutes on a CPU. This is negligible compared to the GPU-hours required for RM training. Consequently, the total training

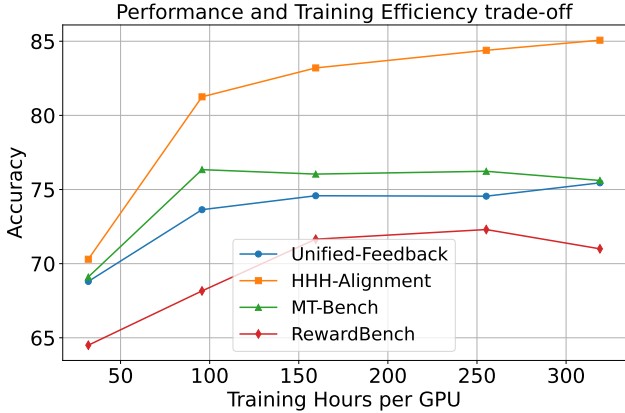

Figure 5: Performance–Training Efficiency trade-off

cost of RMB remains on the same order as existing ensemble methods while delivering substantially higher accuracy and robustness.

**Performance–Training Efficiency Trade-Off** Fig. 5 illustrates the preference prediction accuracy against the total training cost (in GPU-hours per GPU) for varying ensemble sizes $N \in \{1, 3, 5, 8, 10\}$. Several key trends emerge from this analysis. First, we observe monotonic improvements for small $N$; increasing $N$ from 1 to 5, yields consistent accuracy gains across both in-distribution and out-of-distribution evaluation sets, with training costs growing roughly linearly. Second, performance gains begin to saturate and exhibit mild overfitting as $N$ increases from 8 to 10. This plateau suggests that as the input dimension (the vector of $N$ RM scores) grows, an overly flexible aggregator may begin to fit spurious correlations or idiosyncratic noise within the specific collection of RMs rather than learning a generalizable aggregation rule, which suggests an overfitting duo to an over-powerful aggregator model. Finally, the performance curve reveals a wide and relatively flat plateau for $N \in [5, 10]$. This stability indicates that RMB is not sensitive to precise tuning of $N$, as a broad range of values yields similarly strong results without sharp performance distinct drops.

# E    DIVERSITY REGULARIZER SELECTION

In this appendix, we present an additional analysis for comparing the HSIC regularizer with two other widely used diversity-promoting objectives: Negative Correlation Learning (NCL) (Liu & Yao, 1999) and Determinantal Point Processes (DPP) (Taskar, 2013).

We prioritize HSIC as the diversity-prompting regularizer in the reward model learning scenario for two reasons. First, HSIC allows for direct control of statistical dependence by measuring and penalizing the dependence between the margin vectors of different RMs. This is precisely the form of diversity required for effective boosting-based aggregation. Second, the HSIC objective is bounded and scale-stable. The empirical HSIC values are strictly bounded between 0 and 1, ensuring a well-behaved scale when combined with the primary reward-learning objective. In contrast, losses based on DPP or NCL are theoretically unbounded from above, which complicates the selection of trade-off coefficients and can destabilize the optimization process.

We conducted a pilot study (available in our anonymized repository) where we replaced the HSIC term with either DPP or NCL while keeping all other settings identical for $N = 5$ RMs. Table 9 reports the resulting preference prediction accuracies on the Unified-Feedback dataset.

The results indicate that both DPP and NCL yield substantially lower accuracies across all five RMs. Furthermore, they proved more sensitive to hyperparameter tuning due to the unbounded magnitude of their loss terms. By contrast, HSIC facilitates the training of individually superior RMs while maintaining stable optimization dynamics. Based on these empirical findings, we adopt HSIC as the diversity regularizer for RMB and exclude DPP and NCL from the main model design.

# F    NEURAL MODEL VS BOOSTING AGGREGATOR

| Regularizer | RM1 | RM2 | RM3 | RM4 | RM5 |
|---|---|---|---|---|---|
| HSIC | 73.40 | 70.73 | 69.29 | 68.05 | 73.84 |
| DPP | 60.58 | 64.93 | 55.20 | 62.74 | 57.19 |
| NCL | 56.02 | 59.45 | 52.87 | 57.40 | 55.16 |

Table 9: Comparison of different diversity regularizers. We report the accuracy (%) of five individually trained reward models (RM1–RM5) under HSIC, DPP, and NCL regularization. HSIC consistently yields stronger RMs, while DPP and NCL significantly underperform.

Finally, we address the design choice of adopting a tree-based boosting aggregator rather than a neural network aggregator trained end-to-end on RM outputs.

The central design principle of RMB is to leverage boosting, where the aggregator is trained to iteratively correct the residual errors of the current combined model. XGBoost naturally embodies this principle by sequentially adding regression trees that focus on samples where the existing ensemble ranking is incorrect. A neural aggregator trained as a single feed-forward network does not inherently operate on this residual-correction basis; instead, it tends to behave as a monolithic function of the RM scores, which include no advantage over boosted trees.

Furthermore, the outputs of individual RMs function as heterogeneous features, differing in scale, reliability, and domain-specific biases. Tree-based boosting aggregators are well-suited to handle such heterogeneity and remain robust to uninformative or noisy dimensions. This robustness is critical when aggregating multiple RMs, a finding supported by prior work in large-scale ranking systems. In contrast, neural aggregators are often more sensitive to feature scaling and regularization. In our preliminary experiments, we found neural aggregators difficult to tune and unstable, particularly when the RMs were intentionally diversified and therefore not all individually strong.

## G LLM EVALUATION PROMPT TEMPLATE

To evaluate the quality of the text generated by policies trained with different reward signals, we employ the LLM-as-judge paradigm, which has been widely adopted for automated evaluation of generative models (Zheng et al., 2023b; Liu et al., 2024; Fan et al., 2025). In particular, we prompt Deepseek-R1 (DeepSeek-AI et al., 2025) to conduct both pairwise and pointwise assessments of the generated responses. Our prompt templates originate from Zheng et al. (2023b), and are show in Tables 10 and 11. To ensure the fairness and reliability of our pairwise comparisons, we adapt the prompt structure in Fan et al. (2025), which is designed to mitigate ID and position bias (Zheng et al., 2023a; Shen et al., 2023).

## H THE USE OF LARGE LANGUAGE MODELS (LLMS)

Gemini2.5 Comanici et al. (2025) was used in a limited capacity to improve writing quality, including checking grammar and rephrasing certain expressions with better sentence structures. In addition, we use it for formatting LaTeX tables and Matplotlib figures. However, we came up with the research ideas, conducted the analyses, and presented the contents without using AI tools.

## I REPRODUCIBILITY STATEMENT

All code is provided via an anonymized Github Repository, including implementations for data loading, reward model training, and policy optimization. The datasets used are publicly available, and we release the complete set of training hyperparameters. Our evaluation approaches are also publicly available and can be fully reproduced.

Please act as an impartial judge and evaluate the quality of the responses provided by two AI assistants to the user question displayed below. Your evaluation should consider correctness and helpfulness. You will be given a reference answer, assistant A's answer, and assistant B's answer. Your job is to evaluate which assistant's answer is better.

Begin your evaluation by comparing both assistants' answers with the reference answer. Identify and correct any mistakes. Avoid any position biases and ensure that the order in which the responses were presented does not influence your decision. Do not allow the length of the responses to influence your evaluation. Do not favor certain names of the assistants. Be as objective as possible.

Source: **[Source]**
Answer **[ID1]**: **[Answer-A]**
Answer **[ID2]**: **[Answer-B]**

After providing your explanation, output your final verdict by strictly following this format: "[[ID1]]" if assistant A is better, "[[ID2]]" if assistant B is better, and "[[ID-TIE]]" for a tie. Your response should use the format:

Overall Quality: one-sentence comparison and explanation
Preferred: [[Answer ID]]

Table 10: Prompt templates for LLM evaluation on text generated by RLHF optimized policy in §3.2 from a pairwise view. This template originates from Zheng et al. (2023b)'s work, and is adapted according to Fan et al. (2025)'s template to avoid ID and position bias, aiming to provide reliable LLM judges for pairwise comparison.

Please act as an impartial judge and evaluate the quality of the response provided by an AI assistant to the user's question shown below. Your evaluation should consider factors such as helpfulness, relevance, accuracy, depth, creativity, and level of detail in the response. Begin your evaluation with a short explanation. Be as objective as possible. After providing your explanation, please rate the response on a scale of 1 to 10 by strictly following this format: "[[rating]]", for example: "Rating: [[5]]".

**[Question]**
{question}
**[The Start of Assistant's Answer]**
{answer}
**[The End of Assistant's Answer]**

Table 11: Prompt templates for LLM evaluation on text generated by RLHF optimized policy in §3.2 from a pointwise view. This template originates from Zheng et al. (2023b)'s work aiming to provide LLM judges for single answer grading.

