# OpenReview forum: "Reward Model Boosting for RLHF"
_ICLR.cc/2026/Conference — Submitted to ICLR 2026_

### Official Review · Reviewer_urA4 · 2025-10-26

**Soundness:** 2
**Presentation:** 2
**Contribution:** 2
**Rating:** 4
**Confidence:** 3

**Summary:**

The authors present Reward Model Boosting, a novel two-stage approach to improve the robustness of reward models in RLHF and mitigate the reward hacking problem. The method first trains a set of reward models with an HSIC-based diversity-promoting regularizer to encourage learning complementary reward landscape. Subsequently, it learns a gradient-boosted decision tree aggregator to combine the outputs of these diverse RMs into a more accurate and robust final reward signal. The experiments show that RMB outperforms existing methods on both in-distribution and out-of-distribution datasets.

**Strengths:**

1. The paper aims to address reward hacking, a highly relevant problem in RLHF. The motivation is strong and valuable to the community.

2. The experiments are well-designed to specifically validate the mitigation of reward hacking. The results show that RMB maintains consistent improvement, even under noisy conditions.

3. The proposed method consistently outperforms a set of baselines across both in-distribution and out-of-distribution datasets.

**Weaknesses:**

1. It seems that the proposed method introduces substantial training costs from three aspects:
- Training multiple reward models, each potentially as large as the actor model;
- Computing pairwise HSIC values during training
- An additional boosting stage post-training.
2. The core components—ensembling and boosting-based aggregation—are well-established techniques. The paper’s novelty primarily stems from combining existing methods in an engineering-oriented way, rather than introducing a fundamentally new approach, which makes the overall contribution incremental.
3. Lack of sensitivity analysis with respect to hyperparameters or model structure.

**Questions:**

1. The RMB approach significantly increases training costs. Under a fixed computational budget, how does RMB compare to training a single, larger reward model? Does RMB's advantage hold in a budget-constrained scenario?

2. The current experiments are conducted with a 2B-parameter model. Given that the training cost grows super-linearly with model size, how well does the method scale to larger backbones (e.g., 7B or 13B)?

3. While the introduction of HSIC regularization is interesting, could a large value of $\lambda_{HSIC}$ lead to "over-diversification" and thus reduce the accuracy of individual models? It would be helpful to include a sensitivity analysis with respect to $\lambda$.
In addition, is there any specific reasons for the current design of aggregator?

4. The paper reports that performance peaks when the number of reward models is around 8–10. Is there a principled or data-driven method for determining the optimal ensemble size?

---

> ### Author Response · Authors · 2025-11-20
>
> Thank you for highlighting that Our **motivation is strong and valuable to the community**.
>
> >Weakness 1: It seems that the proposed method introduces substantial training costs from three aspects:
> Training multiple reward models, each potentially as large as the actor model;
> Computing pairwise HSIC values during training
> An additional boosting stage post-training.
>
> We would like to clarify that RMB is **highly computationally efficient** when compared to the appropriate baselines (ensemble methods), and even outperforms single large models with fewer total parameters (see Question 1).
>
> * Reward Model training cost: The appropriate comparison is not to a single weak RM, but to ensemble methods required to mitigate reward hacking [1, 2]. Our experiments demonstrate that RMB with $N=3$ outperforms standard ensembles with $N=5$. Thus, RMB achieves stronger performance with lower total computational cost than the competing ensemble baselines.
>
> * Negligible HSIC Overhead: The computation of the HSIC regularizer is a matrix operation whose cost is negligible compared to the forward and backward passes of the reward models during training.
>
> * Negligible boosting Cost: The boosting stage is extremely lightweight. Empirically, training the XGBoost aggregator took <5 minutes on CPU, which is negligible (~0.2%) compared to the 31.9 GPU-hours required to train a single Reward Model.
> Therefore, the total training cost of RMB is comparable to existing ensemble methods, and as we show, our method is significantly more effective.
>
>
>
>
> >Weakness 2: The core components—ensembling and boosting-based aggregation—are well-established techniques. The paper’s novelty primarily stems from combining existing methods in an engineering-oriented way, rather than introducing a fundamentally new approach, which makes the overall contribution incremental.
>
> We assert that our contribution is a **principled architectural innovation**, not merely an engineering combination.
>
> Standard ensembles (averaging) [1, 2] often fail in reward modeling because the individual models learn similar features and share correlated errors. Our novelty lies in solving this systemic failure via a two-step theoretical framework:
>
> (1) Enforced Diversity: We introduce HSIC not just as a "trick," but to alter the loss landscape, forcing models to learn complementary representations of human preference. This solves the "correlated error" problem inherent in standard ensembles.
>
> (2) Residual Learning: Standard ensembles use averaging, which only reduces variance. We learn a non-linear aggregator in the principle of boosting that reduces bias by correcting the residual errors of the ensemble.
>
> Such two-step synergy (learning diverse RMs and aggregating them via residual learning) constitutes a novel framework for the Reward Modeling domain, which effectively mitigates the reward hacking issue as shown in our experiments in Section 3.3.
>
>
>
> >weakness3 Lack of sensitivity analysis with respect to hyperparameters or model structure.
>
>
> Regarding hyperparameter N: We **have conducted an additional hyperparameter analysis wrt N in our new revision**, and the new results are reported in Appendix D. As shown, the performance curves are wide and flat from N=5 to N=10, suggesting that our approach is robust to the choice of N.
>
> Regarding hyperparameter λ_HSICL: We followed the established practice of selecting $\lambda$ to balance main task loss and diversity regularization [3,4]. Following the reviewer’s suggestion, we **conducted an additional λ_HSICL analysis** experiment sweeping $\lambda_{HSIC}$ across orders of magnitude (from 0.001 to 10).
>
> As shown in the table below, our approach remains robust to the selection of $\lambda$. The RMB performance remains stable and effective across a wide range ($0.001 \le \lambda \le 1$), despite the variation in pairwise correlation. Performance only degrades when the regularization becomes overwhelmingly dominant ($\lambda=10$)
>
> | $\lambda_{HSIC}$ | Avg. Indiv. RM Perf. | Avg. Pairwise Corr. | RMB Perf. |
> |------------------|----------------------|----------------------|-----------|
> | 0.001            | 70.7 $\pm$ 0.8        | 92%                  | 71.7      |
> | 0.01             | 71.3 $\pm$ 1.1        | 86%                  | 72.8      |
> | 0.1              | 71.1 $\pm$ 2.5        | 73%                  | 74.6  |
> | 1                | 68.8 $\pm$ 3.0        | 59%                  | 71.2      |
> | 10               | 61.7 $\pm$ 3.7        | 51%                  | 66.5      |
>
>
> More details can be seen in Appendix G in our new revision.
>
> Regarding model structure: Our choice of models was designed to follow the established experimental setup of recent work by [5]. This allowed us to conduct a rigorous, controlled comparison against their reported baselines.

---

> ### Author Response · Authors · 2025-11-20
>
> >Question 1: The RMB approach significantly increases training costs. Under a fixed computational budget, how does RMB compare to training a single, larger reward model? Does RMB's advantage hold in a budget-constrained scenario?
>
> We would like to clarify that our RMB **does not introduce substantial training costs** compared to other established literatures that require training multiple reward models, as explained in weakness 1.
>
> In addition, to directly address the reviewer’s question, we **conducted a new analysis** on the RewardBench dataset. We compared our RMB (aggregating Gemma-2B models) against single reward models that are larger (7B and 13B). As shown in the table below, RMB’s advantage holds in a budget-constrained scenario.
>
> | Method           | Backbone           | Total Params (Proxy for Cost) | RewardBench Score |
> |------------------|--------------------|-------------------------------|-------------------|
> | RewardModel-Mistral | Mistral-7B         | 7B                            | 67.0              |
> | UltraRM          | LLaMA-13B          | 13B                           | 69.8              |
> | RMB    | 3 $\times$ Gemma-2B | 6B                           | 72.0          |
> | RMB  | 5 $\times$ Gemma-2B | 10B                          | 72.2          |
>
>
> >Question 2: The current experiments are conducted with a 2B-parameter model. Given that the training cost grows super-linearly with model size, how well does the method scale to larger backbones (e.g., 7B or 13B)?
>
> Unfortunately, conducting the full set of experiments on 7B+ backbones on the RLHF task was beyond the computational resources available for our lab.
>
> In addition, the core mechanisms of RMB are optimization and ensemble techniques that are theoretically agnostic to the underlying neural architecture size. Since reward hacking is a result of proxy imperfection, the mitigation RMB provides is expected to generalize to larger backbones.
>
>
>
> >Question 3: While the introduction of HSIC regularization is interesting, could a large value of lead to "over-diversification" and thus reduce the accuracy of individual models? It would be helpful to include a sensitivity analysis with respect to . In addition, is there any specific reasons for the current design of aggregator?
>
> On Over-diversification: Please refer to our response to weakness 3.
>
> On Aggregator Design: We chose the boosting aggregator for two reasons:
>
> (1) Residual Learning: Unlike neural aggregators or averaging, boosting explicitly trains the aggregator to identify which model to trust for specific inputs, effectively patching the 'blind spots' of one model with the strengths of another.
>
> (2) Robustness to heterogeneity: Tree-based aggregators are invariant to the scale and distribution of inputs, making them ideal for aggregating outputs from diverse neural networks [6,7]. Our experiments in Section 3 empirically justified the robustness of this selection.
>
>
> >Question 4: The paper reports that performance peaks when the number of reward models is around 8–10. Is there a principled or data-driven method for determining the optimal ensemble size?
>
> Data-driven method: Based on the new analysis we reported in Appendix D, our model is robust to the choice of N. In practice, we recommend treating $N$ as a hyperparameter to be selected via validation on a small hold-out preference set
>
>
> >Summary:
>
> We sincerely thank the reviewer again for their time and feedback. We have aimed to provide thorough responses to each concern and have already run the additional experiments suggested to strengthen our evaluation. We believe these revisions will significantly improve the paper.
>
> In addition, we would be grateful if you could take this additional context into account when revisiting your evaluation.  We are happy to answer any further questions.
>
>
> [1] Coste, et al. 2024. Reward model ensembles help mitigate overoptimization. ICLR.
>
> [2] Eisenstein, et al. 2024. Helping or herding? reward model ensembles mitigate but do not eliminate reward hacking. COLM.
>
> [3] Dong, et al. 2023. Diversity-enhancing Generative Network for Few-shot Hypothesis Adaptation. ICML.
>
> [4] Ma, et al. 2024. Towards continual learning desiderata via hsic-bottleneck orthogonalization and equiangular embedding. AAAI.
>
> [5] Yang, et al. 2024. Regularizing hidden states enables learning generalizable reward model for LLMs. NIPS.
>
> [6] Grinsztajn, et al. 2022. Why do tree-based models still outperform deep learning on tabular data? NIPS.
>
> [7] Shwartz-Zi, et al. 2021. Tabular Data: Deep Learning is Not All You Need. Information Fusion.

---

> > ### Comment · Reviewer_urA4 · 2025-11-25
> >
> > Thank you for the rebuttal. The additional sensitivity analysis provided at the rebuttal stage makes the paper more complete and addresses most of my earlier concerns. I am raising my score to weak accept and recommend incorporating the lambda sensitivity analysis into the main paper.

---

> > > ### Author Response · Authors · 2025-11-25
> > >
> > > We sincerely thank the reviewer for the decision to raise the score. As recommended, we have now incorporated the hyperparameter sensitivity analysis into the main paper, which can be found in Section 3.6.

---

### Official Review · Reviewer_DuEg · 2025-10-29

**Soundness:** 3
**Presentation:** 3
**Contribution:** 3
**Rating:** 6
**Confidence:** 3

**Summary:**

This paper introduces Reward Model Boosting (RMB), a novel approach to improve the robustness and reliability of reward models used in Reinforcement Learning from Human Feedback (RLHF). The authors identify that traditional reward models are prone to reward hacking—where optimizing the proxy reward degrades alignment with true human preference—due to imperfect supervision. RMB mitigates this by (1) training multiple diverse reward models using a Hilbert–Schmidt Independence Criterion (HSIC)-based regularizer (§2.3) to reduce correlation among models, and (2) aggregating them with a boosted decision tree trained in the spirit of XGBoost (§2.2). Extensive experiments across four datasets (Unified-Feedback, HHH-Alignment, MT-Bench, and RewardBench) demonstrate that RMB significantly improves both in-distribution and out-of-distribution (OOD) reward accuracy (§3.2), reduces reward hacking under Best-of-N sampling and PPO training (§3.3), and shows robustness against label noise and text perturbations (§3.5). The results suggest that RMB yields a more trustworthy reward signal for downstream RLHF optimization.

**Strengths:**

Originality: The idea of applying boosting principles to ensemble reward models in RLHF is original and well-motivated. While prior work has explored ensembles for robustness (e.g., Coste et al., 2024), this paper’s use of a gradient boosting aggregator to iteratively minimize residual reward errors (Eqn. (3)) and the addition of HSIC-based diversity regularization (Eqn. (8)) represent a novel synthesis of ideas. The focus on margin diversity rather than raw score diversity (§2.3) is also conceptually sound and innovative.

Quality: The empirical evaluation is sound. Table 1 shows that RMB (N = 3–10) consistently outperforms GRM, Label Smooth, Margin Loss, and Avg Ensemble baselines in preference prediction accuracy. Table 2 and Figure 1–2 further validate robustness in both synthetic and RLHF settings. The ablation in Table 3 and correlation heatmaps in Figure 3 provide convincing evidence that HSIC-induced diversity is key to boosting performance.

Clarity: The paper is clearly structured. Mathematical formulations are precise—especially the detailed boosting objective (Eqns. (3)–(5)) and HSIC margin definition (Eqn. (7)). Figures and tables are well-integrated and support the claims effectively.

Significance: The contributions are significant for the RLHF community. The approach is general and could be integrated into diverse RLHF pipelines. Moreover, the demonstrated OOD generalization and robustness to label noise suggest practical benefits for real-world LLM alignment.

**Weaknesses:**

Dependence on tree-based aggregation: The reliance on decision-tree aggregators (XGBoost) may limit scalability or differentiability for large-scale LLM training. A brief comparison to neural aggregators or soft ensemble weighting would be beneficial.

Diversity regularization tuning: The choice of the HSIC coefficient λ_HSIC is mentioned but not systematically analyzed. Since Table 3 shows performance sensitivity to diversity, an ablation on λ_HSIC or kernel choice would make the results more complete.

Limited exploration of scalability and efficiency: The paper does not discuss training cost or inference latency when increasing N from 3 to 10 reward models (Table 1). Given the growing size of LLM-based reward models, it would be valuable to evaluate computational overhead and potential efficiency trade-offs.

Evaluation scope: The experiments rely on relatively small models (Gemma-2B-IT, Mistral-7B). While this is acceptable for a proof of concept, the practical significance for frontier-scale RLHF systems (tens to hundreds of billions of parameters) remains uncertain.

**Questions:**

Hyperparameter sensitivity: How sensitive is the performance to λ_HSIC and the number of boosting trees K?

Scalability: Could the boosting aggregator be replaced with a neural network trained end-to-end for differentiability within the RLHF loop?

Computational cost: What is the runtime overhead of training N = 10 RMs plus the aggregator compared to a single RM?

---

> ### Author Response · Authors · 2025-11-20
>
> We thank the reviewers for highlighting that our idea is “**conceptually sound and innovative**”, “**empirical evaluation is sound**”, “paper is clearly structured”, and “contributions are significant”.
>
> >Weakness 1: Dependence on tree-based aggregation: ... A brief comparison to neural aggregators or soft ensemble weighting would be beneficial.
>
> Thank you for this suggestion.
>
> Regarding "comparison to soft ensemble": We would like to clarify that we **use this as a primary baseline in all experiments** in Sections 3.2, 3.3, 3.4, and 3.5. Our experimental results consistently demonstrate that our RMB approach significantly outperforms this standard soft ensemble. This confirms that learning a non-linear aggregation strategy (Boosting) provides value beyond simple weighting.
>
> Regarding “comparison to Neural aggregators”: We selected a tree-based approach (XGBoost) instead of a neural aggregator for three reasons: superiority on heterogeneous data, boosting principles, and computational efficiency.
>
> * Superiority on heterogeneous data: The input to the aggregator is a vector of reward scores from different models, which constitutes heterogeneous tabular data. Recent research [0, 1] has demonstrated that tree-based models (like XGBoost) outperform deep learning methods on heterogeneous datasets. Specifically, [1] shows that tree-based models are more robust to uninformative features and irregular decision boundaries compared to Neural Networks, which often struggle with invariant biases on low-dimensional data.
>
> * Alignment with Boosting Principles: Our method is grounded in the principle of gradient boosting: iteratively fitting the residual errors of the ensemble. XGBoost is designed for this formulation. It allows us to correct the weaknesses of the existing ensemble members, while NNs often overfit on such tasks [1].
>
> * Inference Efficiency for RLHF: In the RLHF loop, the reward model is queried frequently to evaluate generated samples. Latency is a critical bottleneck. A decision-tree ensemble is computationally lightweight (simpler CPU instructions) compared to a Neural Network (heavy matrix multiplications). As shown in our Appendix C.2, the inference latency of our tree aggregator is negligible, whereas a neural aggregator would introduce significant overhead, reducing training efficiency for RLHF.
>
> We have put a brief comparison of neural aggregators in Appendix F in our new revision.
>
>
> >Weakness 2: Diversity regularization tuning: The choice of ... λ_HSIC is ... not systematically analyzed. ... an ablation on λ_HSIC or kernel choice ...
>
> Regarding hyperparameter λ_HSICL:  We followed the established practice of selecting $\lambda$ to balance main task loss and diversity regularization [4,5]. Following the reviewer’s suggestion, we **conducted an additional λ_HSICL analysis** experiment sweeping $\lambda_{HSIC}$ across orders of magnitude (from 0.001 to 10).
>
> As shown in the table below, our approach remains robust to the selection of $\lambda$. The RMB performance remains stable and effective across a wide range ($0.001 \le \lambda \le 1$), despite the variation in pairwise correlation. Performance only degrades when the regularization becomes overwhelmingly dominant ($\lambda=10$)
>
> | $\lambda_{HSIC}$ | Avg. Indiv. RM Perf. | Avg. Pairwise Corr. | RMB Perf. |
> |------------------|----------------------|----------------------|-----------|
> | 0.001            | 70.7 $\pm$ 0.8        | 92%                  | 71.7      |
> | 0.01             | 71.3 $\pm$ 1.1        | 86%                  | 72.8      |
> | 0.1              | 71.1 $\pm$ 2.5        | 73%                  | 74.6  |
> | 1                | 68.8 $\pm$ 3.0        | 59%                  | 71.2      |
> | 10               | 61.7 $\pm$ 3.7        | 51%                  | 66.5      |
>
>
> More details can be seen in Appendix G in our new revision.
>
> Regarding kernel selection: we chose the Gaussian RBF kernel as it is **the standard, widely-adopted** choice in recent HSIC-related research [4-8]. This is because it is a **universal kernel** (meaning HSIC can detect any form of dependence) and is **numerically stable and smooth**. While other kernels exist, they often trade away one of these important properties. We will add this justification to the appendix to make our choice more explicit.
>
>
> >Weakness 3: Limited exploration of scalability and efficiency: ... evaluate computational overhead and potential efficiency trade-offs.
>
> Thank you for your suggestion.
>
> For the reward model training efficiency-performance trade-off, we **have added an additional analysis in Appendix D** in our new revision.
>
> Inference latency is discussed and experimented with in Appendix C.2, which concludes that the RMB adds a negligible inference latency (0.000088) to reward signal prediction compared to other established literature, namely, reward model ensemble [2,3], which requires training multiple reward models as well, making its impact on the overall RLHF training time minimal.

---

> ### Author Response · Authors · 2025-11-20
>
> >Weakness 4: Evaluation scope: The experiments rely on relatively small models (Gemma-2B-IT, Mistral-7B). While this is acceptable for a proof of concept, the practical significance for frontier-scale RLHF systems (tens to hundreds of billions of parameters) remains uncertain.
>
> Our choice of models was designed to **follow the established experimental setup** of recent work by [9]. This allowed us to conduct a rigorous, controlled comparison against their reported baselines.
>
> Training frontier-scale models is computationally prohibitive for most academic labs. Our work aims to provide the foundational proof of concept and rigorous empirical evidence on accessible, standard benchmarks. We believe our strong results will inspire and justify follow-up validation by industry labs on models with tens or hundreds of billions of parameters.
>
> >Question 1: Hyperparameter sensitivity: How sensitive is the performance to λ_HSIC and the number of boosting trees K?
>
> Thank you for the chance to clarify this.
>
>
> For λ_HSIC, as discussed in our response to Weakness 2, the selection is principled (main loss dominant) and follows standard practice.
>
> For the number of trees K, our approach is not sensitive to K because K is **not a fixed hyperparameter**. In boosting, the model continues to add trees until a convergence criterion (namely, the residual error falling below a threshold) is met. The residual threshold can be easily set by monitoring performance on a validation set to prevent overfitting, which is a standard procedure.
>
> >Question 2: Scalability: Could the boosting aggregator be replaced with a neural network trained end-to-end for differentiability within the RLHF loop?
>
> Thank you for your question. Please refer to weakness 1.
>
>
> >Question 3: Computational cost: What is the runtime overhead of training N = 10 RMs plus the aggregator compared to a single RM?
>
> The aggregator's own training time is negligible. The cost of training N=10 RMs is, as expected, 10x that of a single RM.
> However, we argue that the relevant comparison is not to a single RM but to other ensemble methods [2, 3] that also require training N models for mitigating the reward hacking issue.
>
> Viewed this way, our method is highly cost-effective. Our experiments show that RMB with N=3 outperforms an average ensemble with N=5. This demonstrates that our approach can achieve a stronger, more robust reward signal with less computational cost than existing ensemble techniques.
>
> In addition, we have provided a more detailed training efficiency analysis in Appendix D in our new revision.
>
>
> >Summary:
>
> Thank you again for your time and for recognizing the scientific value of our work. We have aimed to provide thorough responses to each concern and have already run the additional experiments suggested to strengthen our evaluation. We believe these revisions will significantly improve the paper.
>
> In addition, we would be grateful if you could take this additional context into account when revisiting your evaluation.  We are happy to answer any further questions.
>
>
> [0] Grinsztajn, et al. 2022. Why do tree-based models still outperform deep learning on tabular data? NIPS.
>
> [1] Shwartz-Zi, et al. 2021. Tabular Data: Deep Learning is Not All You Need. Information Fusion.
>
> [2] Coste, et al. 2024. Reward model ensembles help mitigate overoptimization. ICLR.
>
> [3] Eisenstein, et al. 2024. Helping or herding? reward model ensembles mitigate but do not eliminate reward hacking. COLM.
>
> [4] Dong, et al. 2023. Diversity-enhancing Generative Network for Few-shot Hypothesis Adaptation. ICML.
>
> [5] Ma, et al. 2024. Towards continual learning desiderata via hsic-bottleneck orthogonalization and equiangular embedding. AAAI.
>
> [6] Ji et al. 2021. CondenseNet with exclusive lasso regularization. PCM.
>
> [7] Gao et al. 2025. Diversity-induced fuzzy clustering with Laplacian regularization. Information Sciences.
>
> [8] Chen et al. 2023 Explore and Exploit the Diverse Knowledge in Model Zoo for Domain Generalization. ICML.
>
> [9] Yang, et al. 2024. Regularizing hidden states enables learning generalizable reward model for LLMs. NIPS.

---

> > ### Comment · Reviewer_DuEg · 2025-11-25
> >
> > Dear author(s), thank you for engaging in the rebuttal! After reading the reviews, my score remans unchanged.

---

> > > ### Author Response · Authors · 2025-11-25
> > >
> > > Thank you for checking our author response. In case you have any other questions, we're ready to discuss further.

---

### Official Review · Reviewer_mFLE · 2025-11-01

**Soundness:** 3
**Presentation:** 4
**Contribution:** 3
**Rating:** 4
**Confidence:** 4

**Summary:**

The paper proposes a new method called Reward Model Boosting (RMB) to improve the reliability of reward signals in RLHF, in an attempt to address the reward hacking issue. The model itself is a collection of several traditional language reward models, optimized so that their pairwise distance defined from Hilbert–Schmidt Independence Criterion (HSIC) is large. Experiments show the effectiveness of both: using HSIC based distance in training diverse reward models; and using decision tree boosting to improve the final prediction performance.

**Strengths:**

1. It's interesting to incorporate boosting methods in RLHF. Using ensembles has been widely explored before, but boosting looks like a technique that hasn't been used so far.

2. Using HSIC as diversity regularization has strong motivation and clear explanation.

3. Experiments indicate the RMB method is capable of more robust reward modeling than single models / average ensembles.

**Weaknesses:**

1. The paper lacks explanation on how HSIC is superior to other diversity regularization methods. I think there should be ablation on simpler diversity regularizations and HSIC.

2. Some experimental results are not clearly explained. In Table 3, both individual RM and average ensemble benefits from using HSIC than random, which may contain other more deeper reasons why the regularization works better instead of just introducing diversity.

3. This seems an interesting topic overall of applying higher level learning tricks to reward modeling, but the novelty is somewhat compromised by DPO partially replacing RLHF in practice.

**Questions:**

1. Why did you choose HSIC as the diversity regularization? Did you try other methods?

2. Can you provide more explanation on Weakness 2?

3. On Table 1, what is special about reasoning that makes RMB perform worse than marginal loss?

4. With later DPO methods instead of RLHF, improving on general reward modeling / solving reward hacking seems a little marginal overall. This method seems more suitable to be used in digging more specific domains, e.g. security, etc. Do you have any idea how your method can tackle these domains?

---

> ### Author Response · Authors · 2025-11-20
>
> We thank the reviewer for highlighting that our idea is interesting, strongly motivated, and clearly explained.
>
> >Weakness 1: The paper lacks explanation on how HSIC is superior to other diversity regularization methods. I think there should be ablation on simpler diversity regularizations and HSIC.
>
> Thank you for encouraging us to include this analysis.
>
> We chose HSIC because (1) HSIC directly measures and penalizes statistical independence, which is exactly what we need, and (2) HSIC values are bounded between 0 and 1 (as opposed to some other common metrics like DPP [1] and NCL [2]), which is critical for stable optimization.
>
> **In our pilot study, we have already tried DPP [1] and NCL [2] **as the diversity-promoting regularizer (which can be verified in our anonymized repo). As mentioned, DPP and NCL are unbounded, potentially going to infinity. Thus, it can cause difficulty in optimizing the overall objective:
>
> training loss = RM loss + diversity regularizer loss.
>
> The preliminary results of our pilot study showed that DPP and NCL yield subpar performance, so we did not further explore them in our model design.
>
> We’re grateful for the reviewer’s comment, and have included the comparison (summarized below) in our revision.
>
> | | RM1 | RM2 | RM3 | RM4 | RM5 |
>  | :--- | :---: | :---: | :---: | :---: | :---: |
> | HSIC | 73.40 | 70.73 | 69.29 | 68.05 | 73.84 |
> | DPP | 60.58 | 64.93 | 55.20 | 62.74 | 57.19 |
> | NCL | 56.02 | 59.45 | 52.87 | 57.40 | 55.16 |
>
> Thank you for encouraging us to include this analysis. We’ve added the analysis in Appendix E in the revision.
>
>
> >Weakness 2: Some experimental results are not clearly explained. In Table 3, both individual RM and average ensemble benefits from using HSIC than random, which may contain other more deeper reasons why the regularization works better instead of just introducing diversity.
>
> Thank you for this observation.
>
> Individual RMs benefit from using HSIC: We notice that the average RM performance has only increased slightly.  We conducted two-sided t-tests, and their p-values are 66% and 36% (which are >> 5%), respectively. Therefore, they’re **not** statistically significant.
>
> Average ensemble benefits from using HSIC: It is shown that **ensembles (including average ensemble) may benefit from diversity** [3,4,5]. This is because diverse individuals have different strengths and weaknesses. They will **make different errors on different inputs**. Even with simple averaging, the errors may be smoothed out.
>
> That being said, our RBM method **benefits more from the diversity measure** (ours: 3.6% gain, averaging ensemble: 1.2% gain), confirming that our boosting aggregator is more capable than simple averaging at utilizing diverse information captured by HSIC-regularized RMs.
>
>
>
> >Weakness 3: This seems an interesting topic overall of applying higher level learning tricks to reward modeling, but the novelty is somewhat compromised by DPO partially replacing RLHF in practice.
>
> We appreciate this perspective on the evolving alignment landscape. However, we respectfully disagree that DPO compromises the relevance of RLHF for state-of-the-art, large-scale models.
>
> The DPO-like methods do not dominate the post-training or preference alignment. While DPO has gained significant popularity, its adoption has been **most prominent in academic-level fine-tunes and with smaller-scale LMs**. This popularity stems from its simplicity, as it neatly removes the need for a separate reward model and the complex RL exploration process.
>
> RLHF and its variants (like RLVR, GRPO, etc.) **remain central to industry-level post-training for large-scale LMs**. Major labs with substantial RL pipelines (e.g., the creators of the "official" DeepSeek, Qwen, and Llama families) continue to use RLHF for their commercial-level applications.The core advantage of RLHF is generalization via exploration: a trained Reward Model allows the policy to explore and find high-quality trajectories that do not exist in the static preference dataset used by DPO.
>
> Because RLHF remains central to the development of the highest-performing generalist models, improving the robustness of the reward signal (which RMB does) addresses one of the most critical bottlenecks in the current industry-standard pipeline.

---

> ### Author Response · Authors · 2025-11-20
>
> >Question 1: Why did you choose HSIC as the diversity regularization? Did you try other methods?
>
> Thank you for your question. Please refer to the response to weakness 1.
>
> >Question 2: Can you provide more explanation on Weakness 2?
>
> Thank you for your question. Please refer to the response to weakness 2.
>
> >Question 3: On Table 1, what is special about reasoning that makes RMB perform worse than marginal loss?
>
> Thank you for the close reading of our results. We offer two clarifying points:
>
> Overall Performance: **RMB outperforms Margin Loss on 7 out of 8 tasks in Table 1**. Reasoning is the sole exception, and the difference is slight (72.3 vs 72.7).
>
> Nature of the Signal: We hypothesize this is due to the objective nature of reasoning tasks (e.g., Math). In subjective tasks (Creative Writing, Chat), the preference signal is "soft" and noisy, which RMB handles well by aggregating diverse views. In Math, the signal is often binary (Correct/Incorrect). Margin loss produces a "harder" decision boundary that may be marginally better suited for such binary correctness tasks.
>
> RMB's superior robustness across the other 7 tasks highlights its value as a general-purpose solution.
>
>
> >Question 4: With later DPO methods instead of RLHF, improving on general reward modeling / solving reward hacking seems a little marginal overall. This method seems more suitable to be used in digging more specific domains, e.g. security, etc. Do you have any idea how your method can tackle these domains?
>
> Thank you for this question. RMB is suited for specific domains (like security). However, we believe that **RMB is also a good solution for general-purpose, cross-domain alignment**.
>
> **General Alignment is a Multi-Domain Problem**: General-purpose alignment is essentially a collection of heterogeneous domains (e.g., Reasoning, Coding, Creative Writing, Safety). A single monolithic reward model often struggles to balance these conflicting objectives. RMB solves this via the principle of boosting:
>
> * HSIC Regularization: Our HSIC regularizer explicitly forces the ensemble members to capture heterogeneous features in the preference data. In a general-purpose dataset, one RM might focus more on "correctness" features (crucial for Math/Code) while another focuses on "tone" or "safety" features.
>
> * Tree-Based Aggregation: The lightweight boosting aggregator then dynamically selects the most relevant signals for the current input. This allows the system to switch gears effectively across domains, generalizing the domain-specific strength to the cross-domain setting.
>
> Empirical Evidence: As shown in Table 1 and Table 2, **RMB achieves significant performance gains on Out-Of-Distribution (OOD) datasets**. This demonstrates that by capturing heterogeneous features during training, RMB does not overfit to a single domain but instead learns a robust reward signal that generalizes well to unseen tasks, which is a key requirement for general-purpose foundation models.
>
> The Limitations of DPO for cross-domain alignment: Recent literature suggests **DPO struggles with broad generalization**. [6,7] show that DPO is highly sensitive to distribution shifts and often degrades on OOD tasks compared to RLHF. The state-of-the-art generalist models (e.g., Llama 3, DeepSeek-V3) continue to rely on RLHF because its the online exploration provided by a robust reward model is necessary to cover the open-ended space of general user queries. **RMB directly upgrades this critical component**.
>
>
> >Summary:
>
> We sincerely thank the reviewer for reviewing our manuscript so carefully and for providing insightful questions. We have carefully responded to each of the points in detail and hope our clarifications dispel any remaining questions.
>
> The reviewer gave high breakdown scores (soundness 3, presentation 4, and contribution 3), but the initial rating was 4 (below average). We suspect that the reviewer has misclicked the button. It'll be highly appreciated if the reviewer could double-check the rating and update it accordingly.
>
>
> [1] Kulesza, et al. 2012. Determinantal Point Processes for Machine Learning. Foundations and Trends textregistered in Machine Learning.
>
> [2] Liu, et al. 1999. Ensemble learning via negative correlation. Neural Networks.
>
> [3] Wood, et al. 2023. A Unified Theory of Diversity in Ensemble Learning. JMLR.
>
> [4] Wang et al. 2020. Transductive Ensemble Learning for Neural Machine Translation. AAAI.
>
> [5] Shayegh, et al. 2024. Error Diversity Matters: An Error-Resistant Ensemble Method for Unsupervised Dependency Parsing. ACL
>
> [6] Xu, et al. 2024. Is DPO Superior to PPO for LLM Alignment? A Comprehensive Study. ICML.
>
> [7] Ivison, et al. 2024. Unpacking DPO and PPO: Disentangling Best Practices for Learning from Preference Feedback. NIPS.

---

> > ### Comment · Reviewer_mFLE · 2025-11-27
> >
> > Thank the authors for their prompt response. I will raise my score to 6 as the authors answered my questions and addressed my concerns.

---

> > > ### Author Response · Authors · 2025-11-27
> > >
> > > We appreciate your decision to raise the score. We are glad that our response and results have addressed your concerns.

---

### Official Review · Reviewer_c3kk · 2025-11-03

**Soundness:** 2
**Presentation:** 3
**Contribution:** 2
**Rating:** 4
**Confidence:** 2

**Summary:**

This paper introduces Reward Model Boosting (RMB), a novel approach designed to mitigate reward hacking in RLHF. The core components of RMB include a lightweight aggregator based on XGBoost and a diversity-promoting regularizer using the Hilbert-Schmidt Independence Criterion (HSIC). The aggregator operates on the principle of boosting, iteratively learning to combine the outputs of multiple reward models to produce a more accurate and robust final reward signal. Concurrently, RMB trains an ensemble of reward models, encouraging them to evaluate rewards from diverse perspectives. Through extensive experiments, the paper demonstrates that RMB significantly outperforms existing methods in reward accuracy on both in-distribution and out-of-distribution datasets. It also effectively alleviates reward hacking, ultimately enhancing the overall performance of RLHF.

**Strengths:**

1. The problem addressed in this paper—reward hacking—is a well-recognized and critical challenge in the RLHF domain. Any research that can effectively mitigate this issue is of significant value. The paper clearly identifies the root cause of the problem (the imperfection of proxy models) and precisely situates its research direction.

2. The paper's evaluation is comprehensive, conducted across multiple standard datasets covering both in-distribution and OOD scenarios. By comparing against several strong baselines, the authors not only validate the accuracy of their proposed method but also provide robust evidence of its effectiveness in mitigating reward hacking and its strong robustness through a variety of experiments.

**Weaknesses:**

1. Besides HSIC, there are other methods in the literature for promoting model diversity. The paper lacks a comparison with some of these common diversity-promoting techniques. The work would be significantly strengthened by including a discussion on the theoretical advantages of choosing HSIC, supplemented by experimental comparisons with alternative approaches.

2. The RMB method requires the simultaneous training of N reward models, along with an additional HSIC regularization term, which undoubtedly increases training complexity and computational overhead. Although the appendix mentions the required computational resources, the total cost is multiplied compared to training a single RM. It is recommended that the authors provide a more explicit discussion of this overhead in the main text and analyze the trade-off between performance gains and computational costs.

3. As shown in Table 1, the performance of RMB improves as N increases from 3 to 8, but it appears to saturate or slightly decline for some metrics when N=10. This suggests that the choice of N is a critical factor. I would suggest that the authors conduct a more in-depth analysis of the combined impact of N on performance, diversity, and computational cost, and provide practical guidance for selecting an optimal N.

**Questions:**

1. In the experiments, noise is simulated by randomly flipping 25% of the labels, which validates the model's robustness to random errors. However, real-world human feedback often contains systematic biases rather than purely random noise (e.g., a preference for longer, more verbose responses, regardless of information density). How does RMB perform in the presence of such systematic biases? If the training data contains a "length bias" or a "sycophancy bias," can RMB's diversity mechanism identify and mitigate these undesirable tendencies, or would it be misled in the same way as a baseline model?


2. As shown in Table 2, the paper uses DeepSeek-R1 as the judge to evaluate the quality of policies trained with different reward models. While this is a common evaluation paradigm, the judge itself is an LLM with its own inherent biases. Is it possible that the preferences of DeepSeek-R1 happen to align more closely with the style of the model trained via RMB, thus leading to its higher win rate? Have the authors considered validating these results with other judges or a small-scale human evaluation to ensure the conclusion is robust?

---

> ### Author Response · Authors · 2025-11-20
>
> Thank you for saying our work “is of significant value” and that the “evaluation is comprehensive”.
>
> >Weakness 1: Besides HSIC, there are other methods in the literature for promoting model diversity. The paper lacks a comparison with some of these common diversity-promoting techniques. The work would be significantly strengthened by including a discussion on the theoretical advantages of choosing HSIC, supplemented by experimental comparisons with alternative approaches.
>
> We chose HSIC because (1) HSIC directly measures and penalizes statistical independence, which is exactly what we need, and (2) HSIC values are bounded between 0 and 1 (as opposed to some other common metrics like DPP [1] and NCL [2]), which is critical for stable optimization.
>
> **In our pilot study, we have already tried DPP [1] and NCL [2]** as the diversity-promoting regularizer (which can be verified in our anonymized repo). As mentioned, DPP and NCL are unbounded, potentially going to infinity. Thus, it can cause difficulty in optimizing the overall objective:
>
> training loss = RM loss + diversity regularizer loss.
>
> The preliminary results of our pilot study showed that DPP and NCL yield subpar performance, so we did not further explore them in our model design.
>
> We’re grateful for the reviewer’s comment, and **have included the comparison (summarized below) in our revision**.
>
> | | RM1 | RM2 | RM3 | RM4 | RM5 |
>  | :--- | :---: | :---: | :---: | :---: | :---: |
> | HSIC | 73.40 | 70.73 | 69.29 | 68.05 | 73.84 |
> | DPP | 60.58 | 64.93 | 55.20 | 62.74 | 57.19 |
> | NCL | 56.02 | 59.45 | 52.87 | 57.40 | 55.16 |
>
> Thank you for encouraging us to include this analysis. We’ve added this discussion and these tables to Appendix E to provide a more complete picture for the reader in the revision.
>
>
> >Weakness 2: The RMB method requires the simultaneous training of N reward models, along with an additional HSIC regularization term, which undoubtedly increases training complexity and computational overhead. Although the appendix mentions the required computational resources, the total cost is multiplied compared to training a single RM. It is recommended that the authors provide a more explicit discussion of this overhead in the main text and analyze the trade-off between performance gains and computational costs.
>
> Thank you for raising this concern.
>
> We would like to clarify that our **RMB does not introduce substantial training costs** compared with other established methods, namely, reward model ensembles [3,4], and reward model merging [5], which require training multiple reward models as well.
>
> When compared to these relevant baselines, our RMB is highly efficient and effective. Our experiments show that RMB with N=3 outperforms an ensemble method with N=3 and N=5, demonstrating that RMB can achieve stronger performance with less computational cost than competing ensemble methods.
>
> On the other hand, we do not introduce substantial overhead latency for RLHF either. We provide detailed overhead latency analysis in Appendix C.2 (page 19), which concludes that the overhead of RMB is negligible compared with [3,4] because it adds a negligible training latency ratio (0.000088), making its impact on the overall RLHF training time minimal.
>
>
> We revised the manuscript and provided a detailed training efficiency analysis in Appendix D.
>
>
> >Weakness 3 As shown in Table 1, the performance of RMB improves as N increases from 3 to 8, but it appears to saturate or slightly decline for some metrics when N=10. This suggests that the choice of N is a critical factor. I would suggest that the authors conduct a more in-depth analysis of the combined impact of N on performance, diversity, and computational cost, and provide practical guidance for selecting an optimal N.
>
>
> Thank you for this observation about the choice of N. We appreciate the opportunity to clarify our perspective.
>
> Saturate at N=10: The performance saturation from N=8 to N=10 is a sign of overfitting in the boosting aggregator. As N increases, the aggregator's input dimension (the N reward signals) grows. With a large N, the aggregator may begin to fit to spurious correlations or noise specific to the outputs of those N particular base models, rather than learning a generalizable aggregation strategy. This is a standard trade-off of model complexity (overfitting vs. underfitting).
>
> Robustness in selection of N: We conducted additional analysis following the reviewer’s suggestion (thanks!). The new results are reported in Appendix D in our revision. As shown, the curves are wide and flat from N=5 to N=10, suggesting that **our approach is robust to the choice of N**.
>
> Practical Guidance for selecting optimal N: Based on the above analysis, our model is robust to the choice of N, and a moderate N (e.g., 5--10) will just work well. In addition, a user may further optimize the hyperparameter N by validation on a hold-out set.
>
> **We’ve added the discussion in Appendix D**.

---

> ### Author Response · Authors · 2025-11-20
>
> >Question 1: In the experiments, noise is simulated by randomly flipping 25% of the labels, which validates the model's robustness to random errors. However, real-world human feedback often contains systematic biases rather than purely random noise (e.g., a preference for longer, more verbose responses, regardless of information density). How does RMB perform in the presence of such systematic biases? If the training data contains a "length bias" or a "sycophancy bias," can RMB's diversity mechanism identify and mitigate these undesirable tendencies, or would it be misled in the same way as a baseline model?
>
> Thank you for raising this question.
>
> Previous studies show human-preference data typically contains 20-30% label noise [6]. Later, researchers use random flipping to simulate this label noise [7], which reflects the fact that human annotators often do not think about the candidates thoroughly, and may label their preference arbitrarily. **To make a rigorous comparison**, we adopt the same setting as [7] to illustrate the reward hacking mitigation under this label noise setting.
>
> Besides, we find it interesting to distinguish between systematic label bias and reward hacking:
>
> **Systematic Label Bias** is when the labeled data itself is flawed, e.g., reward model learned on such datasets systematically prefers longer responses regardless of quality.
>
> **Reward Hacking** (our focus) is when the labeled data is correct, but the proxy reward model fails to capture the true human preference perfectly, leading the policy to exploit the model's imperfections.
>
> Our RMB approach is designed to solve the second problem. It **learns a more robust reward signal** that better captures the true underlying human preference already present in the training data.
>
> Therefore, RMB may not directly mitigate systematic biases that originate from a flawed data-labeling process. That being said, we posit that RMB's ability to create a more robust reward signal would be complementary to existing debiasing techniques (which often involve data preprocessing or regularized training). While explicitly studying this combination is outside our current scope, we agree that this is a very interesting and important direction for future research.
>
>
>
> >Question 2: As shown in Table 2, the paper uses DeepSeek-R1 as the judge to evaluate the quality of policies .... the judge itself is an LLM with its own inherent biases. Is it possible that the preferences of DeepSeek-R1 happen to align more closely with the style of the model trained via RMB, thus leading to its higher win rate? Have the authors considered validating these results with other judges or a small-scale human evaluation to ensure the conclusion is robust?
>
>
> We thank you for this constructive suggestion.
>
> We carefully choose LLM for LLM-as-judge evaluation from a different family of the backbone generator (DeepSeek vs Gemma) to avoid the issue that LLM for LLM-as-judge aligns more closely with the generator.
>
> In addition, generator policies learned with different reward models are judged by the same LLM, thus the generator–judge correlation applies to both our RMB and competing methods, so the comparison is fair and reasonable.
>
> As suggested by the reviewer, we **have conducted an additional evaluation using ChatGPT-4o as the judge**, following the exact same pairwise win-rate methodology. The results confirm our original findings, showing significant improvement over the baselines:
>
> | Comparison        | Win    | Tie   | Lose   | Improve Ratio |
> |-------------------|--------|-------|--------|----------------|
> | vs Baseline RM    | 59.54  | 5.42  | 35.04  | 24.5           |
> | vs Avg Ensemble   | 53.15  | 7.93  | 38.92  | 14.23          |
> | vs GRM            | 57.38  | 9.37  | 33.25  | 24.13          |
>
>
> > Summary
>
> We sincerely thank the reviewer for their time and feedback. We have aimed to provide thorough responses to each concern and have already run the additional experiments suggested to strengthen our evaluation. We believe these revisions will significantly improve the paper.
>
> In addition, we would be grateful if you could take this additional context into account when revisiting your evaluation.  We are happy to answer any further questions.
>
> [1] Kulesza, et al. 2012. Determinantal Point Processes for Machine Learning. Foundations and Trends textregistered in Machine Learning.
>
> [2] Liu, et al. 1999. Ensemble learning via negative correlation. Neural Networks.
>
> [3] Coste, et al. 2024. Reward model ensembles help mitigate overoptimization. ICLR.
>
> [4] Eisenstein, et al. 2024. Helping or herding? reward model ensembles mitigate but do not eliminate reward hacking. COLM.
>
> [5] Rame, et al. 2024. Warm: On the benefits of weight averaged reward models. arXiv.
>
> [6] Wang, et al. 2024. Secrets of rlhf in large language models part ii: Reward modeling. arXiv.
>
> [7] Yanng, et al. 2024. Regularizing hidden states enables learning generalizable reward model for LLMs. NIPS.

---

### Author Response · Authors · 2025-11-21
**General Response to All Reviewers**

We thank all the reviewers for their thoughtful comments and constructive suggestions. We are encouraged that several reviewers recognized the novelty, effectiveness, and significant contribution of our work.

We are pleased to note that the feedback primarily focused on requests for additional analysis regarding training costs and hyperparameter sensitivity, rather than identifying technical flaws or interpretability issues. In our revision, we have carefully addressed the concerns raised.

Due to page limits, we have included the following additional content in the Appendices:

* Training Efficiency (Appendix D): We provide a detailed analysis demonstrating that our approach does not introduce significant training costs compared to established works.

* Pilot Study (Appendices E & F): We have expanded the details on our pilot study to provide further motivation and insight into our approach.

* Hyperparameter Sensitivity (Section 3.6): New experiments demonstrate that our approach is robust to hyperparameter selection.

We have uploaded the revised manuscript with all changes highlighted in red. We have taken great care to address the specific questions raised by each reviewer in our individual responses. If you have any further questions or suggestions, we would be happy to address them.

Once again, thank you for your valuable feedback, which has helped us strengthen the paper.

Best regards,

The Authors

---

### Author Response · Authors · 2025-11-24
**Reminder for reviewer-author discussion**

Dear Reviewers:

We thank your efforts in reviewing our paper. It has been a while since we provided an initial response. However, we have not seen follow-up comments yet.

In our author response, we've clarified all the points, especially for training efficiency, diversity regularizer selection, and hyperparameter selection. We also provided additional experiments to address the concerns.

It'll be highly appreciated if you could take a look at our responses and let us know if there’re any additional concerns.  We’re happy to discuss further and revise our manuscript accordingly.


Thank you very much,
-Authors

---

### Author Response · Authors · 2025-11-30
**Summary of the discussion**

Dear Chairs,

We thank you for your time and effort in chairing our paper. We would like to provide a summary of our paper and the author--reviewer discussion.

Our paper proposes a novel approach, called Reward Model Boosting, which uses advanced boosting to construct a reliable reward signal for RLHF. Our method effectively mitigates reward hacking and significantly improves policy learning in various tasks.

The review scores are summarized as follows:


| Reviewer | c3kk | mFLE | DuEg | urA4 | Average |
| :--- | :--- | :--- | :--- | :--- | :--- |
| Ratings before rebuttal | 4 | 4 | 6 | 4 | 4.5 |
| Ratings after rebuttal | 4 | **6** | 6 | **6** | 5.5 |
| Confidence Scores | **2** | 4 | 3 | 3 | 3 |

Reviewer Status:
* **Reviewers mFLE & urA4:** Raised scores to **6** and confirmed our rebuttal addressed their concerns.
* **Reviewer DuEg:** Maintained a positive score of **6**.
* **Reviewer c3kk:** Gave a score of 4 with **low confidence (2)**, but unfortunately has not had a chance to participate in the discussion.

The reviewers highlighted several **key strengths** of our work:

* **Clear and Strong Motivation:** (c3kk, mFLE, DuEg, urA4)
* **Novel and Interesting Approach:** (mFLE, DuEg)
* **Comprehensive and Sound Evaluation:** (c3kk, mFLE, DuEg, urA4)
* **Significant Contribution:** (DuEg)
* **Clear Writing Structure:** (DuEg)


**Main Concerns** from reviewers:
1. Why choose HSIC:
    * We have explained the rationale as well as our pilot study results in **Appendix E**.
    * Addressed the concerns of **mFLE** (score raised) and **c3kk**.

2. Training Efficiency:
    * We have provided an analysis in **Appendix D**, demonstrating that we do not introduce significant training costs compared to soft-weight ensemble methods.
    * Addressed the concerns of **urA4** (score raised) and **c3kk**.

3. Hyperparameter $\lambda_{\text{HSIC}}$ Sensitivity:
    * We have provided an analysis in **Section 3.6**, demonstrating that our approach is robust to hyperparameter selection.
    * Addressed the concerns of **urA4** (score raised) and **DuEg**.

4. $N$ trade-off:
    * We have provided a detailed analysis in **Appendix D**, which illustrates the trade-off curve on $N$.
    * Addressed the concerns of **urA4** (score raised) and **c3kk**.

**Other concerns**: We have also provided detailed responses and revisions regarding: extending model size to hundreds of billions of parameters (**DuEg**), a conceptualized comparison of DPO and RLHF (**mFLE**), a discussion on the advantage of HSIC (**mFLE**), and a discussion on the advantage of the boosting method (**DuEg**).

**Revision**: We have uploaded the revised manuscript with all changes highlighted in **red**.

We sincerely hope that the AC could take into consideration our responses and revisions and render a convincing decision.

Thank you very much!

---

### Meta-Review · Area_Chair_YEre · 2025-12-29

**Summary:**

The paper propose reward model boosting (RMB) to address reward hacking, which is an important question in RLHF. RMB trains a set of reward models and incorporate similar idea of XGboost. I second the reviewers' comments that using gradient boost to iteratively boost reward model is an interesting direction. However, the manuscript lacks theoretical guidelines on hyperparameter selections. It also not does provide any intuition about selection of HSIC. I recommend to revise the manuscript for more thorough understanding of the reward boosting to enhance the contribution.

**Reviewer Concerns:**

Unaddressed comments. Reviewer c3kk questioned about N-value saturation and systematic bias mitigation. While the authors argued that performance saturates at $N=10$ due to overfitting in the boosting aggregator, there is no guaranteed guideline on choosing the optimal $N$. Same for systematic bias introduced by label bias. Reviewer c3kk further noted that the use of DeepSeek-R1 as a judge might introduce its own biases. Although the authors added a GPT-4o evaluation to confirm the results, the underlying concern remains.

**Reviewer Scores:**

Reviewers urA4  and mFLE could increase from 4 to 6. Reviewer c3kk will remain 4. Reviewer DuEg will stay 6.

---

### Decision · Program_Chairs · 2026-01-26

Reject